# Rational Tuning of LLM Cascades via Probabilistic Modeling

**Michael J. Zellinger and Matt Thomson**

**Reviewed on OpenReview:** `https://openreview.net/forum?id=YCBVcGSZeR`

## Abstract

Understanding the reliability of large language models (LLMs) has recently garnered significant attention. Given LLMs' propensity to hallucinate, as well as their high sensitivity to prompt design, it is already challenging to predict the performance of an individual LLM. However, the problem becomes more complex for compound LLM systems such as cascades, where in addition to each model's standalone performance, we must understand how the error rates of different models interact. In this paper, we present a probabilistic model for the joint performance distribution of a sequence of LLMs, which enables a framework for rationally tuning the confidence thresholds of a LLM cascade using continuous optimization. Compared to selecting confidence thresholds using Bayesian optimization, our parametric Markov-copula model yields more favorable error-cost trade-offs, improving the area under the error-cost curve by 4.3% on average for cascades with $k \geq 3$ models. In the low-sample regime with $n \leq 30$ training examples, the performance improvement widens to 10.2%, suggesting that our framework's inductive assumptions about the interactions between the error rates of different LLMs enhance sample efficiency. Overall, our Markov-copula model provides a rational basis for tuning LLM cascade performance and points to the potential of probabilistic methods in analyzing systems of LLMs.

## 1 Introduction

As LLMs become workhorses of the modern computing stack, systems of LLMs have received significant attention (Zaharia et al., 2024; Chen et al., 2024b). These approaches make it possible to adapt computational spending to the performance requirements at the query or task level (Kag et al., 2023; Chen et al., 2023), yielding significant gains in operational efficiency. These gains are achievable even when accessing LLMs entirely via black-box API calls, by switching between models of different capabilities.

However, moving from single LLMs to LLM systems introduces significant additional complexity. To find the system's optimal operating point, it is important to understand not just the performance of individual models but also the interactions between their error rates. For example, in a simple two-model LLM cascade in which a small model delegates difficult queries to a large model, the large model's error rate increases conditional on receiving a query, since the small model's confidence gating induces an adverse selection (Zellinger and Thomson, 2024).

In this paper, we present a parametric probabilistic model for the joint distribution of the calibrated confidences of a sequence of $k$ LLMs, providing a rational basis for understanding the performance of LLM cascades. We focus on cascades whose constituent models are ordered by size, from smallest to largest. Our probabilistic model is based on a Markov factorization, leveraging the insight that LLMs similar in size are more predictive of each other's confidence. After using logistic regression to calibrate each LLM's confidence, we account for the pairwise interactions between subsequent LLMs' error rates using bivariate copulas, providing a data-efficient model of cascade performance that performs well with as few as $n \leq 30$ training examples across six benchmarks.

Our Markov-copula model makes it possible to tune the confidence thresholds of an LLM cascade using continuous optimization.

Compared to selecting these thresholds via Bayesian optimization, our Rational Tuning framework yields increasingly better error-cost trade-offs as cascade length grows. For cascades with $k \geq 3$ models, our method improves the area under the error-cost curve by 4.3% on average. Compared to high-resolution grid search, the improvement is 2.0%. At the same time, our algorithm significantly improves runtime scaling compared to grid search. For example, we reduce scaling with respect to the cascade length $k$ from exponential to low-order polynomial, making it much faster to tune longer cascades consisting of $k \geq 5$ models.

Relative to the prior literature on LLM cascades, our main contributions are as follows:

- We propose a generative probabilistic model for the joint distribution of the calibrated confidences of a sequence of LLMs, based on a Markov factorization, copula modeling, and mixed discrete-continuous marginal distributions. We demonstrate that our model fits the empirical data well: on the test sets, we report average Cramér-von Mises statistics of $\sqrt{n}\text{CvM} = 0.006$ for the copula models and $\sqrt{\text{CvM}} = 4\%$ for the mixed discrete-continuous marginal distributions.

- Building on our Markov-copula model, we develop an algorithm for tuning the confidence thresholds of an LLM cascade using continuous optimization, based on an analytic probabilistic model. We demonstrate that as cascade length grows, our method increasingly outperforms the error-cost trade-offs obtained with Bayesian optimization and high-resolution grid search baselines. In addition, relative to grid search our method significantly improves the computational complexity of finding optimal confidence thresholds, turning the dependencies on cascade length and the desired resolution of the error-cost curve from intractable and high-order polynomial into low-order polynomial and linear, respectively.

In addition, we present comprehensive evidence that simple hyperparameter-free feature transforms significantly improve the performance of calibrating LLM confidence with logistic regression (Zellinger and Thomson, 2024), demonstrating a 28.2% average reduction in expected calibration error across 10 LLMs and 6 benchmarks.

## 2 Background and Related Work

**Language Models**: given a predefined token vocabulary $\mathcal{V}$, a large language model (LLM) $M$ defines an autoregressive probability distribution $t \sim p(\cdot|t_1, ..., t_n)$ for the next token $t \in \mathcal{V}$ given a sequence of tokens $(t_1, ..., t_n) \in \mathcal{V}^n$. In this work, we focus on the overall input-output behavior of the model $M$. We let $x$ stand for the entire query consisting of tokens $(t_1, ..., t_n)$ and write $M(x)$ for the sequence of tokens $(t_{n+1}, t_{n+2}, ...)$ obtained when repeatedly sampling $t_{j+1} \sim P(\cdot|t_1, ..., t_j)$ for $j \geq n$ until encountering a stop token $t_\varnothing \in \mathcal{V}$.

**Language Model Cascades**: a length-$k$ LLM cascade $C = M_1 \rightarrow ... \rightarrow M_k$ routes an incoming query $x$ sequentially from model $M_i$ to $M_{i+1}$ based on confidence measures $\Phi_i = \Phi_i(x) \in [0, 1]$. When $x$ reaches $M_i$, the cascade returns $M_i(x)$ if $\Phi_i(x) > \phi_i$, where $\phi_i \in (0, 1)$ is a confidence threshold for model $M_i$. Otherwise, $C$ forwards the query $x$ to the next model, $M_{i+1}$. Writing $C_{i:k}$ for the subcascade $M_i \rightarrow ... \rightarrow M_k$ consisting of the last $k - i + 1$ models, the output $C(x)$ of the overall cascade is defined recursively as

$$C(x) = \begin{cases} M_1(x) & \text{if } \Phi_1(x) > \phi_1 \text{ or } |\mathrm{C}| = 1 \\ C_{2:\mathrm{k}}(x) & \text{otherwise,} \end{cases} \tag{1}$$

where $|C|$ is the length of the cascade, for example $|C_{2:k}| = k - 1$.

Different authors have recently explored LLM cascades. Chen et al. (2023) have shown that it is possible to approach the performance of a large LLM at much lower cost by initially sending queries to a small model; Aggarwal et al. (2024) present a flexible cascading approach based on a POMPD router; Yue et al. (2024) propose LLM cascades specifically for mathematical reasoning benchmarks; and Gupta et al. (2024) consider uncertainty at individual token position within longer generations. While many of these approaches use standard uncertain quantification techniques for LLMs (discussed below), some use trained neural networks

for making the decision of forwarding a query $x$ to the next model. Neural network approaches have the potential to make more finegrained distinctions between the capabilities of different LLMs[1], but may require large amounts ($n > 1000$) of task-specific training data to perform well.

Jitkrittum et al. (2024) discuss the limits of forwarding queries based purely on the confidence level of the current model, proposing to train a cascading decision that takes into account not only the current model's probability of correctness, but also that of the following model. In addition, Wang et al. (2024) explore finetuning LLMs to make them more effective as part of a cascade. Other methods for LLM orchestration use routers that directly forward queries to suitable LLMs in a one-to-many architecture (Ding et al., 2024; Kag et al., 2023; Sakota et al., 2024; Hari and Thomson, 2023). In addition, some work has explored recombining the string outputs of several smaller models to yield improved performance (Jiang et al., 2023).

**Uncertainty Quantification and Calibration**: LLMs within a cascades require the means to tell "easy" queries from "difficult" ones. Several authors have proposed methods for quantifying this uncertainty. These methods work in different ways. Some draw on the LLMs' intrinsic next-token probabilities (Hendrycks and Gimpel, 2018; Plaut et al., 2024), while others use prompting to elicit confidence statements (Lin et al., 2022a; Kadavath et al., 2022; Xiong et al., 2024). Some sample repeatedly from the LLM and measure the consistency between different answers (Wang et al., 2023; Manakul et al., 2023; Farquhar et al., 2024; Lin et al., 2024), while others train lightweight probes on top of an LLM's hidden embeddings (Azaria and Mitchell, 2023; Ren et al., 2023, Chen et al., 2024a; Kossen et al., 2024). Finally, it is even possible to evaluate uncertainty in a largely unsupervised way (Burns et al., 2024).

*Calibration* of LLM uncertainty refers to the question of whether numerical confidence scores reflect the true probabilities of error. Methods relying on LLMs' next-token probabilities face the challenge that these probabilities are typically overconfident, at least for instruction-tuned LLMs (Ouyang et al., 2022; OpenAI, 2024). Although calibration is not required for forwarding queries based on confidence, it is important for accurately predicting error rates and desirable for gaining insights into system performance. Many techniques for calibration have been proposed (Platt, 1999; Zadrozny and Elkan, 2002; Naeini et al., 2015, Guo et al., 2017; Jiang et al., 2021). Temperature scaling, which divides an LLM's log probabilities by a suitable constant factor (typically $>1$), is often favored for its simplicity.

**Copula Models**: copula models are statistical tools for modeling joint probability distributions. They are widely used in applications. For example, in quantitative finance they are used to price complex securities such as baskets of loans whose repayments depend on multiple borrowers. Mathematically, a *copula* is a joint cumulative distribution function whose marginals all follow the uniform distribution. The idea behind copula modeling is that, in order to specify an arbitrary joint distribution $p(x, y)$, it suffices to specify the marginals $p(x)$, $p(y)$ along with a copula accounting for the correlation between $x$ and $y$. This result is known as

**Theorem 1** (Sklar's Theorem). *Let $F_{XY}$ be a joint distribution function with marginals $F_X$ and $F_Y$. Then there exists a copula $C$ such that for all $x, y \in \mathbb{R}$,*

$$F_{XY}(x, y) = C(F_X(x), F_Y(y)). \qquad (2)$$

*Conversely, if $C$ is a copula and $F_X$ and $F_Y$ are distribution functions, then the distribution function $F_{XY}$ defined by (2) is a joint distribution function with marginals $F_X$ and $F_Y$.*

For a proof and further discussion, see Nelsen (2006). Intuitively, copula modeling builds on the probability integral transform principle: if $X$ is a continuous random variable with distribution function $F_X(\cdot)$, then $F_X(X)$ follows a uniform distribution (Casella and Berger, 2002). In our application to LLM cascades, we model the joint probability $p(\phi_{i-1}, \phi_i)$ of the calibrated confidences of models $M_i$ and $M_{i-1}$ using a Gumbel copula. This copula depends on a single correlation parameter $\theta$, which can be easily calculated from the rank correlation (Kendall's $\tau$) of the two variables.

---

[1]Of particular interest is the potential for detecting rare cases when a small model correctly answers a query on which a larger model fails.

## 3  Rational Tuning of LLM Cascades via Probabilistic Modeling

### 3.1  Markov-Copula Model

Our probabilistic model for the joint distribution of LLM confidences is based on calibrated confidence scores. We use logistic regression to transform a raw confidence signal $p_{\text{raw}}$ into the calibrated confidence score

$$\phi = \Phi_\theta(p_{\text{raw}}), \tag{3}$$

where $\theta$ are the parameters of the logistic regression. The calibrated confidence $\phi$ estimates the model's probability of correctness based on the raw confidence signal $p_{\text{raw}}$. We calibrate each model separately, resulting in functions $\Phi_1, ..., \Phi_k$ for the models $M_1, ..., M_k$ of a cascade $M_1 \rightarrow ... \rightarrow M_k$. See Section 4.1 for more details. Since the confidence signal $p_{\text{raw}}$ depends on the query $x$, we also write $\phi = \Phi(x)$ in a slight abuse of notation.

Our probabilistic model for the joint distribution of the calibrated confidences $\Phi_1(x), ..., \Phi_k(x)$ consists of three parts. First, we model the marginal distribution of the calibrated confidence of each individual LLM in the cascade. Second, we model the correlation between the calibrated confidences $\Phi_i(x), \Phi_{i+1}(x)$ of adjacent models using copulas. Finally, we construct the full joint distribution by combining the conditional probabilities $p(\phi_{i+1}|\phi_i)$ using the Markov property.

Specifically, given a cascade $M_1 \rightarrow ... \rightarrow M_k$ with trained confidence calibrators $\Phi_1, ..., \Phi_k$, we first fit parametric univariate distributions $F_i(\phi_i|\theta_i)$ to model the true marginal distributions $\mathbb{P}(\Phi_i \leq \phi_i)$. Second, we account for the correlation between adjacent models by fitting copulas $C_{i,i+1}(\cdot, \cdot)$. Each copula $C_{i,i+1}$ makes it possible to compute the joint distribution $F_{i,i+1}(\cdot, \cdot)$ of $(\Phi_i, \Phi_{i+1})$ via

$$F_{i,i+1}(\phi_i, \phi_{i+1}) = C_{ij}(F_i(\phi_i), F_j(\phi_j)), \tag{4}$$

by Theorem 1. Finally, we estimate joint probabilities $\mathbb{P}(\Phi_1 \leq \phi_1, \Phi_2 \leq \phi_2, ..., \Phi_k \leq \phi_k)$ by relying on the Markov assumption

$$\mathbb{P}(\Phi_i \leq \phi_i | \Phi_{i-1} \leq \phi_{i-1}, \Phi_{i-2} \leq \phi_{i-2}, ..., \Phi_1 \leq t_1) \approx \mathbb{P}(\Phi_i \leq \phi_i | \Phi_{i-1} \leq \phi_{i-1}), \tag{5}$$

which implies

$$\mathbb{P}(\Phi_1 \leq \phi_1, \Phi_2 \leq \phi_2, ..., \Phi_i \leq \phi_i) \approx \mathbb{P}(\Phi_1 \leq \phi_1) \prod_{j=2}^{i} \mathbb{P}(\Phi_j \leq \phi_j | \Phi_{j-1} \leq \phi_{j-1}). \tag{6}$$

for any $i = 2, ..., k$ and $\phi_1, ..., \phi_i \in (0, 1)$. We study the validity of assumption (5) in Section 4.3.

### 3.2  Parameter Inference for the Probabilistic Model

In this section, we describe in detail the components of our parametric probabilistic model and how we infer their parameters.

**Continuous-discrete mixture of scaled beta distributions**: to model the marginals of calibrated confidence, we must account for the possibility that LLMs sometimes return perfect confidence $p_{\text{raw}} = 1.0$, possibly as a result of performance optimizations such as quantization (Dettmers et al., 2024; Proskurina et al., 2024). Depending on the LLM and the task, almost half of all queries may return perfect confidence, as is the case of GPT-4o Mini on the MMLU validation set (45.7%).

To accommodate the resulting discrete probability masses at the minimum and maximum calibrated confidence values $\phi_{\text{min}}$ and $\phi_{\text{max}}$, we use a mixed continuous-discrete distribution based on a mixture of two beta distributions. Specifically, we use the distribution function

$$F(\phi|w_1, w_2, \phi_{\text{min}}, \phi_{\text{max}}; \pi, \alpha_1, \beta_1, \alpha_2, \beta_2) = w_{\text{min}}\delta_{\phi_{\text{min}}}(\phi) + w_{\text{max}}\delta_{\phi_{\text{max}}}(\phi) + (1 - w_{\text{min}} - w_{\text{max}})F_{\text{mixture}}(\phi), \tag{7}$$

where $\delta_z$ is the distribution of a point mass at $z$, and $F_{\text{mixture}}(\phi)$ is

$$F_{\text{mixture}}(\phi|\phi_{\min}, \phi_{\max}; \alpha_1, \beta_1; \alpha_2, \beta_2) = \pi F_\beta(\frac{\phi - \phi_{\min}}{\phi_{\max} - \phi_{\min}}|\alpha_1, \beta_1) + (1 - \pi)F_\beta(\frac{\phi - \phi_{\min}}{\phi_{\max} - \phi_{\min}}|\alpha_2, \beta_2). \quad (8)$$

Here, $F_\beta(\cdot|\alpha, \beta)$ is the beta distribution with pdf $f_\beta(x|\alpha, \beta) = x^{\alpha-1}(1 - x)^{\beta-1}$ for $x \in (0, 1)$.

We infer the parameters of the model (7) as follows. First, we estimate the minimum and maximum calibrated confidences $\phi_{\min}$ and $\phi_{\max}$ by their observed minimum and maximum values on the training set. We estimate the corresponding discrete probability masses $w_{\min}$ and $w_{\max}$ by simple counting. Finally, to estimate the mixture of beta distributions (8), we use the expectation-maximization algorithm (Dempster et al., 1977).

**Gumbel copula**: to model the correlations between the calibrated confidences of pairs of LLMs, we use the Gumbel copula $C_\theta(u, v)$ given by

$$C_\theta(u, v) = \exp\left(-\left(\log\left(\frac{1}{u}\right)^\theta + \log\left(\frac{1}{v}\right)^\theta\right)^{\frac{1}{\theta}}\right),. \quad (9)$$

where $\theta > 1$ measures the degree of correlation between $u$ and $v$. To fit $\theta$ from empirical data, we use the relationship

$$\theta = \frac{1}{1 - \tau}, \quad (10)$$

where $\tau$ is Kendall's rank correlation coefficient (Nelsen, 2006).

### 3.3 Tuning the Confidence Thresholds

The purpose of the Markov model (6) is to obtain optimal error-cost tradeoffs for an LLM cascade $C$ by tuning the confidence thresholds. We formulate the optimization problem

$$\boldsymbol{\theta}^* = \arg\min_{\boldsymbol{\theta}} (1 - \mathbb{P}_{\boldsymbol{\theta}}(\text{Correct})) + \lambda \, \mathbb{E}_{\boldsymbol{\theta}}[\text{Cost}], \quad (11)$$

where $\boldsymbol{\theta} \in \mathbb{R}^{k-1}$ denotes the confidence thresholds $(\phi_1, ..., \phi_{k-1})$. The Lagrange multiplier $\lambda \geq 0$ indicates the user's cost sensitivity. Setting $\lambda = 0$ means that cost is irrelevant, whereas $\lambda > 0$ penalizes the use of expensive models. To compute the efficient frontier of optimal $(\mathbb{P}(\text{Correct}), \mathbb{E}[\text{Cost}])$ tuples, we solve (11) for different values of the cost sensitivity $\lambda$.

Since $\lambda$ has no known relationship with the expected cost, it is not clear how to choose $\lambda$ to obtain uniform coverage of the efficient frontier. In practice, we start with very small values of $\lambda$ and set

$$\lambda \leftarrow (1 + r)\lambda, \quad (12)$$

for some $r > 0$, until the cost constraint is stringent enough to make the expected cost equal to the least expensive model's expected cost. Typically, setting $r$ between 0.25 and 1 performs well. For any potential gaps in coverage, we adaptively interpolate the optimal thresholds. Specifically, if $\lambda^{(i)} < \lambda^{(i+1)}$ yield optimal thresholds $\boldsymbol{\theta}^{(i)}$ and $\boldsymbol{\theta}^{(i+1)}$ and the gap $|\boldsymbol{\theta}_j^{(i+1)} - \boldsymbol{\theta}_j^{(i)}| = |\phi_j^{(i+1)} - \phi_j^{(i)}|$ for any individual threshold exceeds probability mass $q$ based on the distribution of the calibrated confidence $\Phi_j$, we insert

$$\boldsymbol{\theta}^{(i+1/2)} = (\boldsymbol{\theta}^{(i)} + \boldsymbol{\theta}^{(i+1)})/2 \quad (13)$$

into the list of optimal thresholds between $\boldsymbol{\theta}^{(i)}$ and $\boldsymbol{\theta}^{(i+1)}$. We repeat the infilling procedure (13) until no gaps remain at level $q$. We have found $q < 0.2$ to perform well.

**Efficient computation and optimization of the objective**: solving the minimization problem (11) requires computing a cascade's probability of correctness and expected cost for candidate confidence thresholds $\boldsymbol{\theta} = (\phi_1, ..., \phi_{k-1}) \in \mathbb{R}^{k-1}$. To compute these quantities, we rely on the decompositions (14) and (15) presented in

**Proposition 2.** *Consider a cascade $M_1 \to ... \to M_k$ with confidence thresholds $(\phi_1, ..., \phi_{k-1})$. Assume that the distribution functions for the calibrated confidences $\Phi_i$ satisfy (5), for $i = 1, 2, ..., k$. Assume further that the expected numbers of input and output tokens, $T_i^{(in)}$ and $T_i^{(out)}$, for each model $i$ are independent of the calibrated confidences $\Phi_1, ..., \Phi_k$. Then the probability of correctness $\mathbb{P}(Correct)$ and expected cost $\mathbb{E}[Cost]$ for the cascade are*

$$\mathbb{P}(Correct) = \int_{\{\Phi_1 > \phi_1\}} \Phi_1(\omega) \; d\mathbb{P}(\omega) \tag{14}$$

$$+ \sum_{i=2}^{k} \mathbb{P}(\Phi_1 \le \phi_1) \left( \prod_{j=2}^{i-1} \mathbb{P}(\Phi_j \le \phi_j | \Phi_{j-1} \le \phi_{j-1}) \right) \int_{\{\Phi_i > \phi_i\}} \Phi_i(\omega) \; d\mathbb{P}(\omega | \Phi_{i-1} \le \phi_{i-1})$$

$$\mathbb{E}[Cost] = (1 - \mathbb{P}(\Phi_1 \le \phi_1)) \; \mathbb{E}[C_1] \tag{15}$$

$$+ \sum_{i=2}^{k} \mathbb{P}(\Phi_1 \le \phi_1) \left( \prod_{j=2}^{i-1} \mathbb{P}(\Phi_j \le \phi_j | \Phi_{j-1} \le \phi_{j-1}) \right) (1 - \mathbb{P}(\Phi_i \le \phi_i | \Phi_{i-1} \le \phi_{i-1})) \sum_{j=1}^{i} \mathbb{E}[C_j],$$

*where $C_i$ is the cost per query of model $i$. Specifically, if $\gamma_i^{(in)}$ and $\gamma_i^{(out)}$ are the costs per input and output token, $C_i = \gamma_i^{(in)} T_i^{(in)} + \gamma_i^{(out)} T_i^{(out)}$. To simplify the notation, we let $\phi_k := -\infty$ (although there is no confidence threshold for the final model in the cascade).*

*Proof.* See Appendix A for a proof. $\square$

By leveraging the structure of the summands in Proposition 2, we efficiently compute (14) and (15) in $O(k)$ time, where $k$ is the length of the cascade. See Appendix B for the algorithm. To solve the minimization problem (11), we use the L-BFGS-B optimizer, a low-memory version of the Broyden–Fletcher–Goldfarb–Shanno algorithm (Liu and Nocedal, 1989) modified to handle simple box constraints.

**Smoothness of the objective**: although our Markov-copula model uses mixed discrete-continuous marginals, the objective (11) is smooth because we restrict each threshold $\phi$ to vary only inside the interior of the interval $(\phi_{\min}, \phi_{\max})$, where the marginal distributions of calibrated confidence are smooth. Leaving out the boundary $\{\phi_{\min}, \phi_{\max}\}$ results in no loss of generality because selecting $\phi \in \{\phi_{min}, \phi_{max}\}$ is equivalent to dropping the model from the cascade (if $\phi = \phi_{\max}$) or dropping all subsequent models (if $\phi = \phi_{\min}$). Within our framework, it is possible to carry out such model selection by evaluating subcascades. After fitting copula models for all pairs of models (rather than only adjacent pairs), evaluating subcascades involves little computational overhead.

## 4 Results

### 4.1 Methodology

**Forwarding Queries**: the models in our cascades decide whether to forward queries by thresholding the calibrated confidence $\phi = \Phi(p_{\text{raw}})$, where $p_{\text{raw}}$ is the raw confidence signal. We obtain $p_{\text{raw}}$ from the model-intrinsic next-token probabilities. On multiple-choice tasks, we take the maximum probability among the answer choices (Hendrycks and Gimpel, 2018; Plaut et al., 2024). In the natural language generation case, we first generate the answer, then send a follow up verification prompt to the model asking "Is the proposed answer <answer> true? Answer only Y or N." We use the probability of the Y token as the confidence signal $p_{\text{raw}}$. Our prompt templates are available in Appendix C.

Since we focus on providing techniques compatible with black-box LLM inference via third-party APIs, we leave consideration of hidden layer-based confidence signals to future work. In addition, we do not consider resampling methods such as semantic entropy (Farquhar et al., 2024). Such methods are compatible with black-box inference, but in the context of LLM cascades, their computational overhead appears prohibitive. For example, at the time of writing inference of Llama3.1 405B typically costs 15 times more than inference

of Llama3.1 8B. In this case, it is likely preferable to directly run the 405B model once rather than forward a query based on $k \approx 15$ resamples of the 8B model. See Appendix D for a table listing small-large model pairings from Meta, Anthropic, and OpenAI, along with their price differentials.

**Confidence Calibration**: raw token probabilities of instruction-tuned LLMs are typically poorly calibrated (Ouyang et al., 2022; Brown et al., 2020; OpenAI, 2024; Plaut et al., 2024). However, calibration is important for accurate error prediction. To obtain calibrated confidence scores, we use logistic regression. We favor this approach over temperature scaling since it yields $p$ values and other statistical metrics that are useful for diagnosing calibration issues, especially in a low-data scenario.

Unfortunately, the overconfidence of the raw token probabilities makes the distribution of raw confidence signals highly peaked. The raw token probabilities accumulate near 1.0, making tiny changes in confidence (for example, $p_{\text{raw}} = 0.98$ vs $p_{\text{raw}} = 0.99$) highly consequential. To enhance the calibration performance of logistic regression, as a pre-processing step we apply hyperparameter-free feature transformations that spread out the overconfident probabilities via asymptotes near $p_{\text{raw}} = 0.0$ and $p_{\text{raw}} = 1.0$. Following Zellinger and Thomson (2024), on multiple-choice tasks we use the transformation

$$\xi(p_{\text{raw}}) = \log \left( \frac{1}{1 - p_{\text{raw}}} \right), \tag{16}$$

whereas on natural language generation tasks, we use

$$\xi(p_{\text{raw}}) = \begin{cases} \log \left( \frac{1}{1 - p_{\text{raw}}} \right) & \text{if } p \geq \frac{1}{2}, \\ \log \left( \frac{1}{p_{\text{raw}}} \right) & \text{if } p < \frac{1}{2}. \end{cases} \tag{17}$$

Importantly, these feature transformations do not require any hyperparameter tuning.

Unfortunately, models sometimes return perfect certainty $p_{\text{raw}} = 1.0$ or $p_{\text{raw}} = 0.0$, making (16) and (17) blow up. To address this problem, we reassign all observations with infinite $\xi$ to the maximum of the finite values of $\xi$. In other words, we define

$$\xi_{\max} = \max\{\xi(p_{\text{raw}}) : (p_{\text{raw}}, y) \in \mathcal{D}, p_{\text{raw}} < \infty\}, \tag{18}$$

where $\mathcal{D}$ is the training set consisting of pairs $(p_{\text{raw}}, y)$, and $y \in \{0, 1\}$ indicates correctness of the model's answer.[2] We set all observations where $\xi = \infty$ to $\xi_{\max}$, and treat $\xi_{\min}$ analogously.

**Benchmarks**: we evaluate our probabilistic model and the error-cost curves of LLM cascades on six language modeling benchmarks including MMLU (Hendrycks et al., 2021); MedMCQA (Pal et al., 2022); TriviaQA (Joshi et al., 2017); XSum (Narayan et al., 2018); GSM8K (Cobbe et al., 2021); and TruthfulQA (Lin et al., 2022b). These tasks include general-purpose knowledge and reasoning, domain-specific QA, open-domain QA, summarization, mathematical reasoning, and truthfulness in the face of adversarially chosen questions.

For each benchmark, we use 300 examples for training, and 1000 examples for testing, except on MMLU and TruthfulQA. On MMLU, the dev set contains only 285 examples, of which we use all. The validation set consists of 1531 examples and is divided into different subjects; to avoid bias from subject selection, we take all 1531 validation examples for testing. On TruthfulQA, the entire data set consists only of 817 observations, of which we randomly select 300 for training and the remaining 517 for testing.

Importantly, we run each benchmark in a **zero-shot** manner, since we believe this setting faithfully reflects off-the-shelf use of LLMs in practice. Appendix C gives the prompt templates we used for each benchmark. To conveniently transform and calibrate the raw confidence scores, track the numbers of input and output tokens, and monitor cost, we ran our evaluations using a preliminary version of the `niagara` Python package for LLM cascading. Code for reproducing the results of the paper is available on GitHub.[3]

---

[2] Note we do not require knowing the *actual* answer of a model, only whether it was correct.

[3] Code for reproducing the results of the paper is available at `github.com/mzelling/rational-llm-cascades`.

Table 1: Overall performance of language models across tasks, evaluated on the $n \approx 1000$ test sets. %Corr is the percentage of correct answers, %ECE is the expected calibration error (after training on the $n \approx 300$ training sets), and %Cert is the percentage of queries for which a model returns log probabilities indicating certainty ($-\infty$ or 0.0).

| | MMLU | | | MedMCQA | | | TriviaQA | | | XSum | | | GSM8K | | | TruthfulQA | | |
|---|---|---|---|---|---|---|---|---|---|---|---|---|---|---|---|---|---|---|
| Model | %Corr | %ECE | %Cert | %Corr | %ECE | %Cert | %Corr | %ECE | %Cert | %Corr | %ECE | %Cert | %Corr | %ECE | %Cert | %Corr | %ECE | %Cert |
| llama3.2-1b | 42.5 | 3.8 | 0.0 | 34.5 | 8.7 | 0.0 | 37.2 | 5.8 | 0.0 | 9.4 | 2.9 | 0.0 | 45.9 | 13.1 | 0.0 | 35.8 | 4.3 | 0.0 |
| llama3.2-3b | 57.2 | 4.0 | 0.0 | 53.1 | 6.8 | 0.0 | 63.3 | 4.5 | 0.0 | 21.2 | 3.6 | 0.0 | 79.2 | 9.5 | 0.0 | 43.3 | 7.5 | 0.0 |
| llama3.1-8b | 63.4 | 4.1 | 0.0 | 51.8 | 9.4 | 0.0 | 78.7 | 6.2 | 0.0 | 50.8 | 3.5 | 0.0 | 84.3 | 4.7 | 0.0 | 50.3 | 7.3 | 0.0 |
| llama3.1-70b | 81.5 | **2.4** | 0.0 | 72.6 | 9.9 | 0.0 | 92.8 | 2.3 | 0.0 | 84.5 | 6.0 | 0.0 | 94.9 | 2.9 | 0.0 | 59.4 | 5.7 | 0.0 |
| llama3.1-405b | **85.2** | 2.9 | 0.1 | 75.7 | 10.8 | 0.0 | 94.9 | 3.0 | 0.1 | 83.9 | 5.4 | 0.0 | **97.1** | 1.9 | 0.5 | 69.2 | 5.6 | 0.0 |
| qwen2.5-32b-c | 75.3 | 5.3 | 0.0 | 55.9 | 6.2 | 0.0 | 70.2 | 8.9 | 0.0 | 69.3 | 4.3 | 0.0 | 95.1 | 3.2 | 0.0 | 57.4 | 5.9 | 0.0 |
| qwen2.5-72b | 82.0 | 4.9 | 0.0 | 69.1 | 7.0 | 0.0 | 87.6 | 3.2 | 0.3 | 95.2 | 2.2 | 15.5 | 95.4 | **1.2** | 79.4 | 57.8 | 7.7 | 0.6 |
| gpt-4o-mini | 74.9 | 4.7 | 45.7 | 66.0 | 5.3 | 27.8 | 90.0 | 2.8 | 76.2 | 97.6 | 2.6 | 38.1 | 92.9 | 3.5 | 48.3 | 59.4 | 7.0 | 26.7 |
| gpt-4o | 83.6 | 4.8 | 22.7 | **76.5** | 2.8 | 4.8 | 96.2 | 2.1 | 0.9 | **99.0** | 0.7 | 0.0 | 95.9 | 2.1 | 4.3 | **72.1** | 3.8 | 0.2 |
| Average | 71.7 | 4.1 | 7.6 | 61.7 | 7.4 | 3.6 | 79.0 | 4.2 | 8.3 | 67.9 | 3.5 | 6.0 | 86.7 | 4.7 | 14.7 | 56.1 | 5.9 | 3.0 |

Table 2: Expected calibration error for logistic regression-based calibration, with (%ECE) and without (%ECE_TF) applying the nonlinear transformations (16) and (17) as a pre-processing step. All values are computed on the $n \approx 1000$ test sets, after fitting the logistic regressions on the $n \approx 300$ training sets. For each benchmark, bold font indicates the better performance. The column %Δ shows the reduction in ECE when using the transformations.

| | MMLU | | | MedMCQA | | | TriviaQA | | | XSum | | | GSM8K | | | TruthfulQA | | |
|---|---|---|---|---|---|---|---|---|---|---|---|---|---|---|---|---|---|---|
| Model | %ECE | %ECE_TF | %Δ | %ECE | %ECE_TF | %Δ | %ECE | %ECE_TF | %Δ | %ECE | %ECE_TF | %Δ | %ECE | %ECE_TF | %Δ | %ECE | %ECE_TF | %Δ |
| llama3.2-1b | **3.8** | 6.1 | -37.7 | **8.7** | 9.8 | -11.2 | **5.8** | 5.8 | 0.0 | 2.9 | **2.7** | 7.4 | **13.1** | 13.2 | -0.8 | 4.3 | 4.3 | 0.0 |
| llama3.2-3b | **4.0** | 7.4 | -45.9 | **6.8** | 10.0 | -32.0 | **4.5** | 14.9 | -69.8 | **3.6** | 3.7 | -2.7 | 9.5 | 9.5 | 0.0 | 7.5 | **6.4** | 17.2 |
| llama3.1-8b | **4.1** | 7.0 | -41.4 | **9.4** | 14.1 | -33.1 | **6.2** | 15.1 | -58.9 | **3.5** | 9.7 | -63.9 | **4.7** | 5.1 | -7.8 | 7.3 | 7.3 | 0.0 |
| llama3.1-70b | **2.4** | 7.9 | -69.6 | **9.9** | 12.5 | -20.8 | **2.3** | 5.1 | -54.9 | **6.0** | 10.4 | -42.3 | **2.9** | 4.5 | -35.6 | 5.7 | **5.3** | 7.5 |
| llama3.1-405b | **2.9** | 10.4 | -72.1 | **10.8** | 14.2 | -24.0 | **3.0** | 4.9 | -38.8 | **5.4** | 10.2 | -47.1 | **1.9** | 3.6 | -47.2 | **5.6** | 10.8 | -48.1 |
| qwen2.5-32b-c | **5.3** | 13.9 | -61.9 | **6.2** | 14.5 | -57.2 | **10.0** | 15.7 | -36.3 | **4.3** | 10.2 | -57.8 | **3.2** | 4.7 | -31.9 | **5.9** | 10.6 | -44.3 |
| qwen2.5-72b | **4.9** | 10.9 | -55.0 | **7.0** | 16.1 | -56.5 | **4.0** | 9.4 | -57.4 | **2.2** | 3.2 | -31.3 | **1.2** | 4.9 | -75.5 | **7.5** | 9.3 | -19.4 |
| gpt-4o-mini | **4.7** | 14.8 | -68.2 | **5.3** | 15.5 | -65.8 | **1.4** | 3.8 | -63.2 | 2.6 | **2.4** | 8.3 | **3.5** | 6.1 | -42.6 | **5.3** | 5.8 | -8.6 |
| gpt-4o | **4.8** | 11.2 | -57.1 | **3.1** | 12.6 | -75.4 | **2.1** | 3.6 | -41.7 | **0.7** | 1.0 | -29.6 | **2.1** | 4.6 | -54.3 | **3.8** | 11.4 | -66.7 |
| Average | 5.0 | 7.9 | -36.7 | 7.6 | 9.8 | -22.5 | 4.5 | 8.2 | -45.0 | 3.1 | 4.3 | -28.2 | 6.2 | 7.1 | -12.7 | 5.1 | 6.7 | -23.9 |

**Evaluation**: to evaluate whether a model's answer is correct on open-ended questions, we use Anthropic's Claude 3.5 Sonnet model as a judge. Note that this judging task is relatively easy since the open-ended benchmarks provide reference answers. For example, on TruthfulQA, we include in the evaluation prompt for Claude a list of correct and incorrect reference answers, as provided by the authors of the benchmark (Lin et al., 2022b). On XSum, we do not use the one-line reference summaries and instead follow G-Eval (Liu et al., 2023) to evaluate a proposed summary in terms of its coherence, consistency, fluency, and relevance (Kryściński et al., 2019). We ask Claude to score each dimension on a scale of 1-5. We consider a summary to be correct if it attains a perfect score (5) in each dimension.

**Language Models**: we work with models from Meta's Llama3 series (1B-405B), Alibaba's Qwen series (Qwen2.5 32B Coder and Qwen 72B), and OpenAI's GPT models (GPT-4o Mini and GPT-4o). All models are instruction-tuned. We used the OpenAI API to run inference with GPT-4o Mini and GPT-4o, and the Fireworks API for all other models.

## 4.2 Performance Summary

Tables 1 and 2 show the overall performance of all the language models across tasks, including the calibration performance. We measure calibration in terms of the expected calibration error (ECE), which we compute adaptively by bucketing confidence scores into 10 bins based on the deciles of their distributions. Tables 1 and 2 yield several interesting findings.

First, some of the models often return raw log probabilities indicating certainty ($-\infty$ or 1.0). This tendency varies strongly by model family. OpenAI's GPT models are especially prone to certainty: on MMLU, for example, GPT-4o Mini returns raw confidence 1.0 on 45.7% of queries, while GPT-4o does so on 22.7% of queries. By contrast, Llama3.1 405B returns perfect confidence only on 0.1% of queries.

Second, the test ECE for our calibration scheme varies by model and by benchmark. The benchmark yielding the poorest calibration is MedMCQA, giving an average test ECE of 7.4% across models. However, some

models give exceptional calibration performance across benchmarks. GPT-4o stands out: its test ECE never exceeds 4.8%, which is its ECE on MMLU.

Overall, we observe that our calibration scheme performs satisfactorily across benchmarks and models, with most benchmarks reporting an average test ECE below 5%. Table 2 ablates the importance of the hyperparameter-free feature transforms (16) and (17) for obtaining effective calibration. Applying these transformations results in much lower test ECE scores, reducing them by 28.2% on average. Figure 10 in Appendix E further verifies calibration by showing that, across models and benchmarks, rejecting queries for which the calibrated confidence is $< 1 - q$ approximately lowers the test error rates to $< q$.

### 4.3 Goodness-of-Fit of the Markov-Copula Model

In this section, we show that our probabilistic model fits the empirical data well. We start by presenting evidence that the Markov assumption (5) approximately holds. Second, we show that our Gumbel copula models successfully account for correlations between the error rates of different LLMs, as measured by low square-rooted Cramér-von Mises (CvM) statistics and low rejection rates of the null hypothesis. Finally, we show that our mixed discrete-continuous mixtures of beta distributions provide an adequate model for the marginal confidence distributions, as measured by low square-rooted CvM scores. However, the high rejection rates of the null hypothesis suggest the potential for further improvements.

#### 4.3.1 Verifying the Markov Assumption

To verify that (5) approximately holds, we first visualize the rank correlation between the calibrated confidences of different models. Figure 1 shows that the Kendall's $\tau$ rank correlation is higher for models of similar sizes. In addition, models sharing the same architectural family (Llama, GPT, or Qwen) are more highly correlated than models of different families.

Our findings suggest that a cascade composed only of Llama models (1B-405B) satisfies the Markov assumption more exactly. Consider Figure 1a as an example. For the Llama cascade, Kendall's $\tau$ is highest near the heatmap's diagonal, suggesting a Markov property. By contrast, the mixed cascade composed of Llama, GPT, and Qwen models shows a more haphazard pattern. For example, the rank correlation between GPT-4o Mini and GPT-4o ($\tau = 0.55$) is higher than that between GPT-4o and Llama3 405B ($\tau = 0.54$), even though the latter pair of models are more similar in size. Similarly, Llama3 405B is more strongly correlated with Llama3 70B ($\tau = 0.58$) than with Qwen2.5 72B ($\tau = 0.46$), even though the latter models are of near-identical size. These examples highlight that, in order for the Markov property to hold based on model size, it seems important that models share the same architectural family.

In Appendix F, we further verify the rank correlation patterns between different LLMs by recomputing the rank correlations only on those queries where both models answer correctly or incorrectly.

To probe the Markov property for the Llama cascade in a different way, we train logistic regressions for predicting correctness of the 8B, 70B, and 405B models based on the calibrated confidences of two ancestor models in the cascade. Specifically, we consider the immediate predecessor model (the Markov predictor) paired with each available earlier ancestor. If the Markov property holds, the Markov predictor should hold much greater significance than any other ancestor. Table 3 lists the results, revealing a diagonal pattern for each benchmark that confirms that the Markov predictor is usually much more significant. However, the earlier ancestor often shares statistical significance. To evaluate the significance of this finding, we also computed the magnitude of the regression coefficients corresponding to Table 3. The coefficients follow a similar pattern, revealing that even if multiple predictors are significant, the Markov predictor usually carries the greatest weight.

In sum, our findings suggest that for cascades composed of models sharing the same architectural family, a Markov property holds approximately, though not exactly.

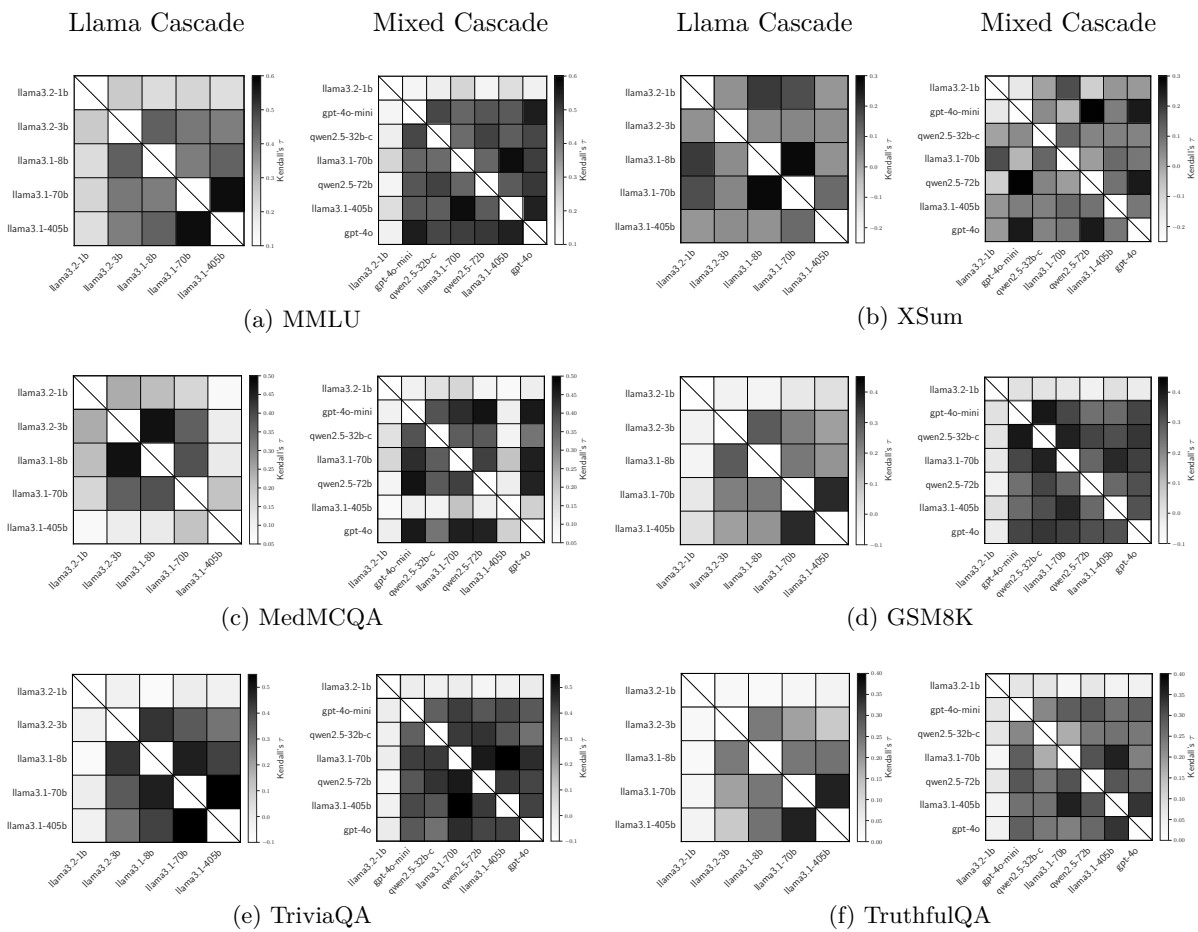

Figure 1: Evaluates the Markov property by showing the Kendall's $\tau$ rank correlation between the calibrated confidences of pairs of LLMs, as evaluated on the test set ($n \approx 1000$ examples). In a Markov pattern, the largest rank correlations occur near the diagonal, based on similarity in model size. For each benchmark, the figure compares the rank correlation structure of a cascade of Llama models to that of a mixed cascade consisting of models from the Llama, GPT, and Qwen families, suggesting that a cascade drawn from models of the same architectural family is more nearly Markovian.

### 4.3.2 Testing the Gumbel Copulas for Modeling LLM Correlations

To evaluate the goodness-of-fit of our Gumbel copula models, we first visualize the correlation between the calibrated confidences of pairs of LLMs. Figures 2 and 3 show scatterplots for several pairs of Qwen, OpenAI, and Llama models. Each scatterplot shows the copula-transformed variables

$$u = \hat{F}_n(\phi), \tag{19}$$

where $\phi$ is the calibrated confidence and $\hat{F}_n$ its empirical distribution on the test set. The marginal distribution of each $u$ is uniform, since we restrict our copula models to the region $(\phi_{\min}, (\phi_{\max})$ of calibrated confidence where the marginal confidence distribution is smooth. Note that Figure 3 highlights the Markov property by showing the increasing rank correlation between Llama models of similar sizes.

We formally test the goodness-of-fit between the fitted Gumbel copulas and the test data by carrying out a Cramér-von Mises test using parametric bootstrapping, following the "Kendall's transform" approach described in Genest et al. (2009). The test involves computing the univariate distribution of copula values $C_{ij}(F_i(x), F_j(x))$ for $x \sim p(x)$, using both the empirical copula and the fitted Gumbel copula. We evaluate the difference between these two distributions using the Cramér-von Mises ($\sqrt{n}$CvM) statistic and obtain a

Table 3: Verifies the Markov property for the Llama cascade by showing the results of using logistic regression to predict each model's correctness based on the calibrated confidences of two ancestor models in the cascade: the immediate predecessor model (Markov predictor) and each available earlier ancestor. For the Markov predictors, the table displays the average $p$ values across all these logistic regressions; for the earlier ancestors, the $p$ value corresponds to a single logistic regression. Underlined values indicate statistical significance (5% level); the lowest $p$ values in each row are bolded. The diagonal pattern in the table suggests the Markov property.

| Benchmark | Predicted | $\log_{10} p$ Value of Markov Predictor vs Earlier Ancestor | | | |
|---|---|---|---|---|---|
| | | 1B | 3B | 8B | 70B |
| MMLU | 8B | -2.66 | **-26.86** | – | – |
| | 70B | -0.52 | -3.48 | **-13.71** | – |
| | 405B | -0.78 | -2.41 | -6.32 | **-25.78** |
| MedMCQA | 8B | -1.85 | **-26.40** | – | – |
| | 70B | -0.26 | -2.72 | **-4.35** | – |
| | 405B | -0.23 | -0.82 | -2.45 | **-24.63** |
| TriviaQA | 8B | -0.14 | **-22.38** | – | – |
| | 70B | -0.58 | -1.02 | **-6.42** | – |
| | 405B | -0.26 | -1.88 | -3.72 | **-11.45** |
| XSum | 8B | -0.72 | **-1.58** | – | – |
| | 70B | -0.97 | -0.61 | **-6.94** | – |
| | 405B | -0.56 | -0.50 | **-2.81** | -1.62 |
| GSM8K | 8B | -2.85 | **-7.48** | – | – |
| | 70B | -0.51 | -0.17 | **-6.49** | – |
| | 405B | -0.36 | -0.13 | **-3.22** | -2.76 |
| TruthfulQA | 8B | **-1.77** | -0.42 | – | – |
| | 70B | -0.30 | -0.44 | **-0.52** | – |
| | 405B | -0.20 | -0.67 | -0.59 | **-1.55** |

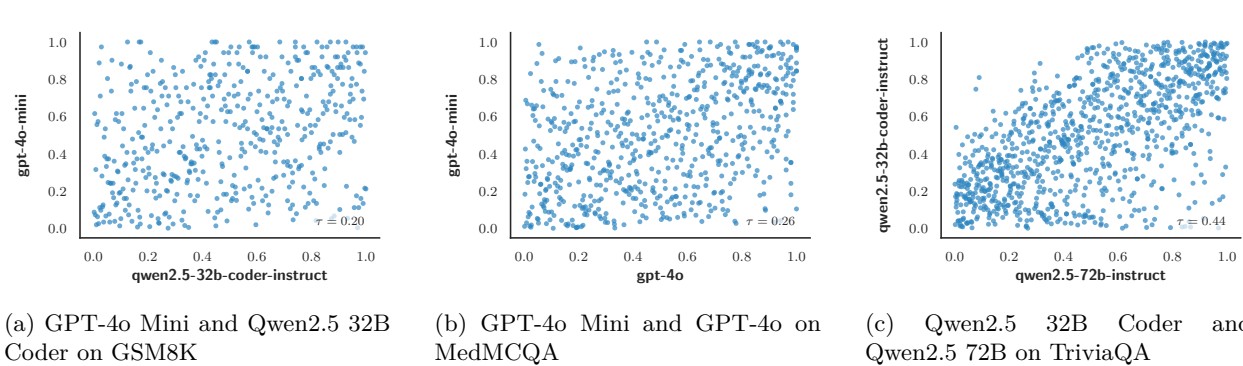

(a) GPT-4o Mini and Qwen2.5 32B Coder on GSM8K

(b) GPT-4o Mini and GPT-4o on MedMCQA

(c) Qwen2.5 32B Coder and Qwen2.5 72B on TriviaQA

Figure 2: Correlations between the calibrated confidences of selected pairs of LLMs on different benchmarks, showing a range of rank correlations between models. The Kendall's $\tau$ rank correlation, shown in the bottom right corner, ranges from $\tau = 0.20$ to $\tau = 0.44$.

$p$ value by parametric bootstrapping with $B = 1000$ samples. In each case, we fit the Gumbel copula on the training data ($n \approx 300$) and evaluate the $p$ value relative to the test data ($n \approx 1000$).

Table 4 breaks down the results by benchmark for two groups of models (Llama models vs OpenAI & Qwen models). Each reported number is based on considering all pairs of models within each group, regardless of similarities in size. There are 10 pairs of Llama models and 6 pairs of Qwen and OpenAI models. The results show that for the Llama models, the fitted Gumbel copulas closely match the empirical correlation

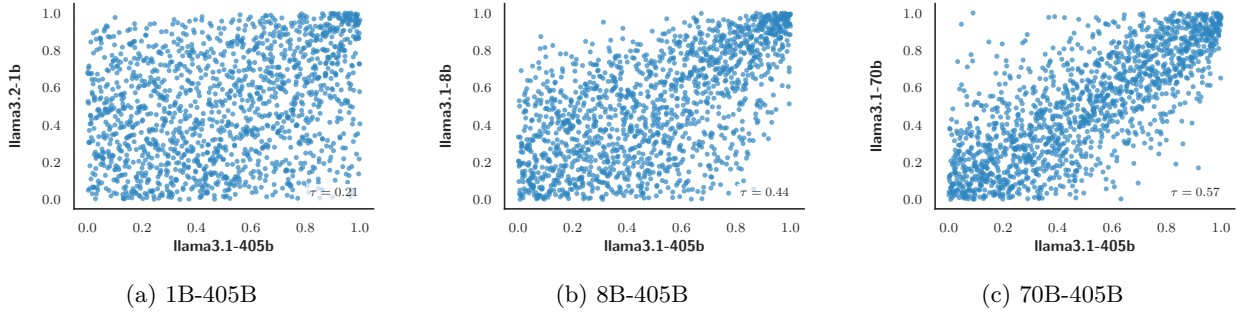

(a) 1B-405B          (b) 8B-405B          (c) 70B-405B

Figure 3: Correlations between the calibrated confidences of smaller Llama models (1B, 8B, 70B) ($y$ axis) and the 405B model ($x$ axis) on MMLU. The increasing rank correlation suggests a Markov property based on model size. The Kendall's $\tau$ rank correlation, shown in the bottom right corner, increases from $\tau = 0.21$ to $\tau = 0.57$.

Table 4: Shows the goodness-of-fit of our Gumbel copula models for modeling pairwise correlations between LLMs, based on a Cramér-von Mises ($\sqrt{n}$CvM) test using parametric bootstrapping. We report the $\sqrt{n}$CvM value, the number of null hypothesis rejections (out of 10 and 6 model pairs for the Llama and Qwen & OpenAI groups, respectively), the percentage of rejections, as well as the geometric and arithmetic mean of $p$ values.

| | Llama Models | | | | | Qwen & OpenAI Models | | | | |
|---|---|---|---|---|---|---|---|---|---|---|
| **Benchmark** | $\sqrt{n}$CvM | #Rej. | % Rej. | $(\prod p)^{\frac{1}{n}}$ | $\frac{1}{n}\sum p$ | $\sqrt{n}$CvM | #Rej. | % Rej. | $(\prod p)^{\frac{1}{n}}$ | $\frac{1}{n}\sum p$ |
| **MMLU** | 0.002 | 0 | 0.0 | 0.569 | 0.591 | 0.011 | 4 | 66.7 | 0.058 | 0.121 |
| **MedMCQA** | 0.004 | 1 | 10.0 | 0.394 | 0.560 | 0.004 | 0 | 0.0 | 0.397 | 0.444 |
| **TriviaQA** | 0.002 | 0 | 0.0 | 0.638 | 0.709 | 0.012 | 2 | 33.3 | 0.078 | 0.187 |
| **XSum** | 0.004 | 0 | 0.0 | 0.405 | 0.480 | 0.002 | 0 | 0.0 | 0.704 | 0.733 |
| **GSM8K** | 0.002 | 0 | 0.0 | 0.688 | 0.757 | 0.016 | 2 | 33.3 | 0.032 | 0.157 |
| **TruthfulQA** | 0.001 | 0 | 0.0 | 0.961 | 0.963 | 0.002 | 0 | 0.0 | 0.800 | 0.812 |
| **Average** | 0.003 | 0 | 1.7 | 0.609 | 0.677 | 0.008 | 1 | 22.2 | 0.345 | 0.409 |

structures between pairs of models on the test set, since the overall rejection rate of the null hypothesis is only 1.7%, well below the 5% rejection rate expected by chance. In addition, the $\sqrt{n}$CvM statistic is only 0.003 on average.

For the group of Qwen and OpenAI models, we observe higher rejection rates. The overall rejection rate of 22.2% suggests that the Gumbel copula model does not fit the data exactly. However, the average $\sqrt{n}$CvM value of 0.008 suggests that the fit is adequate.

### 4.3.3 Testing the Discrete-Continuous Marginal Confidence Distributions

First, we visualize the agreement between the fitted continuous-discrete mixtures of scaled beta distributions and the histograms of calibrated confidence values on the test set. To construct these plots, we first train the calibrators and marginal distributions on the training set ($n \approx 300$ examples).[4] We then compute the calibrated confidence on the test set ($n \approx 1000$) using the trained calibrators. Figure 4 suggests that the fitted marginals align well with the calibrated confidence values on the test data.

Each histogram displays the discrete masses $\phi_{\min}$ and $\phi_{\max}$ of the fitted marginal distributions by shading corresponding areas on the first and last bars of each histogram. We observe in Figure 4a that the discrete probability masses are especially pronounced for GPT-4o Mini on TruthfulQA and GPT-4o on MedMCQA. The trend that the OpenAI GPT models often report certainty also holds for other benchmarks, as Table 1 shows.

---

[4]We do not consider it necessary to train the calibrators and the marginal confidence distributions on separate training data sets, since the calibrators model $p(y|x)$ and the marginal distributions model $p(x)$.

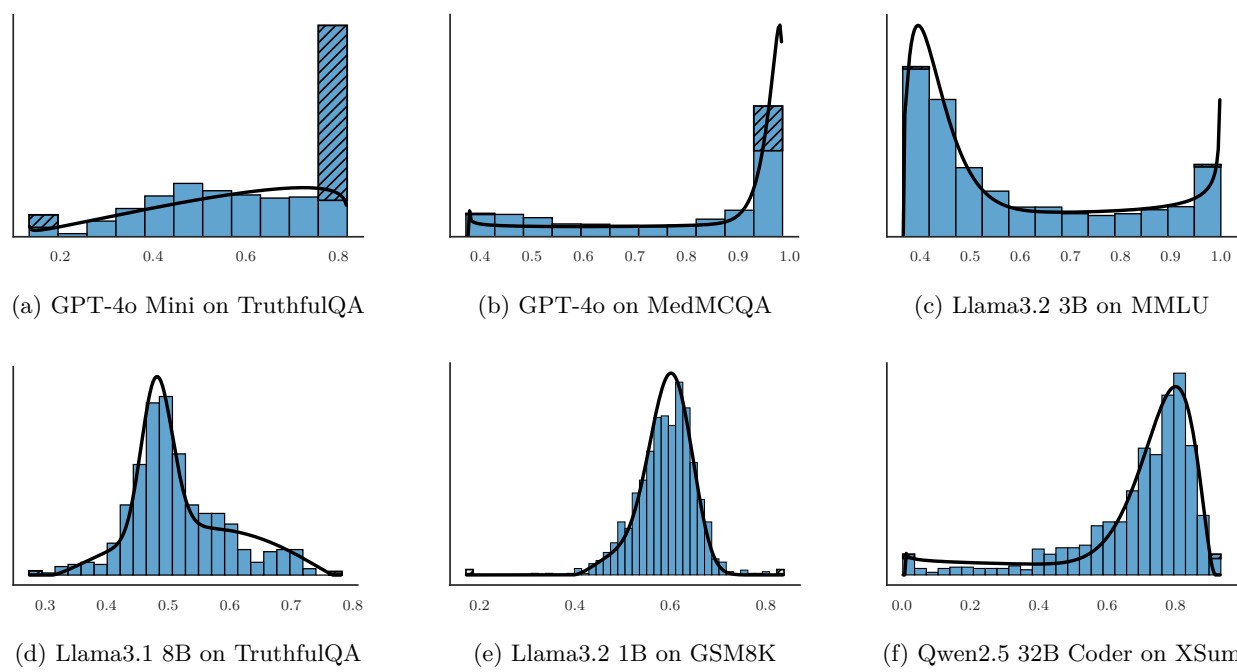

(a) GPT-4o Mini on TruthfulQA    (b) GPT-4o on MedMCQA    (c) Llama3.2 3B on MMLU

(d) Llama3.1 8B on TruthfulQA    (e) Llama3.2 1B on GSM8K    (f) Qwen2.5 32B Coder on XSum

Figure 4: Selection of trained marginal distributions (fitted on $n \approx 300$ training data), with histograms of the test data ($n \approx 1000$). Histogram areas shaded with hatch patterns (especially in (a) and (b)) indicate the contributions of discrete probability masses in our models.

Table 5: Shows the goodness-of-fit of our discrete-continuous mixtures of scaled beta distributions for modeling the marginal distributions of calibrated LLM confidence. We computed $p$ values for the square-rooted Cramér-von Mises ($\sqrt{\text{CvM}}$) statistic using parametric bootstrapping with $B = 1000$ samples. The $\sqrt{\text{CvM}}$ statistic and its $p$ value were computed on the test set ($n \approx 1000$), whereas the marginal distributions were fitted on the training set ($n \approx 300$). We highlight $p < 0.05$ with an underline and $p < 0.001$ with **bold** font. Additionally, we **bold** the largest $\sqrt{\text{CvM}}$ value within each column. Highlighted values indicate the greatest discrepancies with our model.

| Model | MMLU $\sqrt{\text{CvM}}$ | $p$ | MedMCQA $\sqrt{\text{CvM}}$ | $p$ | TriviaQA $\sqrt{\text{CvM}}$ | $p$ | XSum $\sqrt{\text{CvM}}$ | $p$ | GSM8K $\sqrt{\text{CvM}}$ | $p$ | TruthfulQA $\sqrt{\text{CvM}}$ | $p$ |
|---|---|---|---|---|---|---|---|---|---|---|---|---|
| **llama3.2-1b** | 0.031 | **0.000** | 0.025 | 0.015 | 0.018 | 0.117 | 0.036 | 0.001 | 0.026 | 0.015 | 0.025 | 0.109 |
| **llama3.2-3b** | 0.014 | 0.144 | **0.115** | **0.000** | 0.043 | **0.000** | 0.020 | 0.076 | 0.020 | 0.071 | 0.030 | 0.053 |
| **llama3.1-8b** | 0.016 | 0.066 | 0.088 | **0.000** | 0.022 | 0.033 | 0.037 | **0.000** | 0.016 | 0.163 | 0.022 | 0.181 |
| **llama3.1-70b** | **0.048** | **0.000** | **0.137** | **0.000** | **0.057** | **0.000** | **0.070** | **0.000** | 0.038 | **0.000** | 0.044 | 0.002 |
| **llama3.1-405b** | 0.024 | 0.004 | 0.113 | **0.000** | 0.028 | 0.008 | 0.027 | 0.009 | 0.034 | 0.001 | 0.036 | 0.019 |
| **gpt-4o-mini** | 0.032 | **0.000** | 0.060 | **0.000** | 0.008 | 0.441 | 0.020 | 0.077 | 0.026 | 0.016 | 0.028 | 0.072 |
| **qwen2.5-32b-c** | 0.036 | **0.000** | 0.069 | **0.000** | 0.040 | **0.000** | 0.020 | 0.067 | 0.028 | 0.010 | 0.023 | 0.160 |
| **qwen2.5-72b** | 0.028 | 0.001 | 0.073 | **0.000** | 0.040 | **0.000** | 0.041 | **0.000** | 0.004 | 0.678 | 0.036 | 0.018 |
| **gpt-4o** | 0.029 | **0.000** | 0.100 | **0.000** | 0.046 | **0.000** | 0.026 | 0.013 | 0.036 | 0.001 | **0.065** | **0.000** |
| **Average** | 0.029 | 0.024 | 0.087 | 0.003 | 0.034 | 0.066 | 0.033 | 0.027 | 0.025 | 0.106 | 0.034 | 0.068 |

We formally test the goodness-of-fit of the marginal distributions by computing the square-rooted Cramér-von Mises statistic

$$\sqrt{\text{CvM}} = \sqrt{\int (\hat{F}_n^{\text{test}}(x) - F(x|\hat{\boldsymbol{\theta}}))^2 \, dF(x|\hat{\boldsymbol{\theta}})}, \tag{20}$$

where $\hat{F}_n^{\text{test}} = \frac{1}{n} \sum_{i=1}^{n} \delta_{\Phi(x_i)}$ is the empirical distribution of the calibrated confidence on the test data, and $F(\cdot|\boldsymbol{\theta})$ is our marginal distribution model (7) with $\boldsymbol{\theta} = (\phi_{\min}, \phi_{\max}, w_{\min}, w_{\max}, \pi, \alpha_1, \beta_1, \alpha_2, \beta_2)$. In Tables 5 and 6, we report (20) both for $\hat{\boldsymbol{\theta}}$ estimated from the training data ($\sqrt{\text{CvM}}$), and for $\hat{\boldsymbol{\theta}}$ re-fitted on the test

Table 6: Shows the goodness-of-fit of our discrete-continuous mixtures of scaled beta distributions for modeling the marginal distributions of calibrated LLM confidence, after re-fitting the marginal distributions on the test set. We computed $p$ values for the square-rooted Cramér-von Mises ($\sqrt{\mathrm{CvM_r}}$) statistic using parametric bootstrapping with $B = 1000$ samples. We highlight $p < 0.05$ with an underline and $p < 0.001$ with **bold** font. Additionally, we **bold** the largest $\sqrt{\mathrm{CvM_r}}$ value within each column. Highlighted values indicate the greatest discrepancies with our model.

| Model | MMLU | | MedMCQA | | TriviaQA | | XSum | | GSM8K | | TruthfulQA | |
|---|---|---|---|---|---|---|---|---|---|---|---|---|
| | $\sqrt{\mathrm{CvM_r}}$ | $p$ | $\sqrt{\mathrm{CvM_r}}$ | $p$ | $\sqrt{\mathrm{CvM_r}}$ | $p$ | $\sqrt{\mathrm{CvM_r}}$ | $p$ | $\sqrt{\mathrm{CvM_r}}$ | $p$ | $\sqrt{\mathrm{CvM_r}}$ | $p$ |
| **llama3.2-1b** | 0.018 | 0.046 | 0.020 | 0.085 | 0.005 | 0.986 | 0.006 | 0.935 | 0.018 | 0.126 | 0.008 | 0.970 |
| **llama3.2-3b** | 0.009 | 0.461 | 0.010 | 0.608 | 0.036 | **0.000** | 0.009 | 0.666 | 0.010 | 0.566 | 0.013 | 0.661 |
| **llama3.1-8b** | 0.012 | 0.248 | 0.010 | 0.574 | 0.011 | 0.492 | 0.009 | 0.688 | 0.006 | 0.947 | 0.010 | 0.846 |
| **llama3.1-70b** | 0.016 | 0.072 | 0.018 | 0.121 | 0.024 | 0.029 | 0.017 | 0.141 | 0.023 | 0.037 | 0.015 | 0.460 |
| **llama3.1-405b** | 0.015 | 0.133 | 0.011 | 0.498 | 0.017 | 0.130 | 0.006 | 0.933 | 0.031 | 0.002 | 0.019 | 0.290 |
| **gpt-4o-mini** | 0.004 | 0.928 | 0.007 | 0.853 | 0.003 | 0.913 | 0.011 | 0.393 | 0.009 | 0.549 | 0.014 | 0.548 |
| **qwen2.5-32b-c** | 0.011 | 0.282 | 0.012 | 0.355 | 0.020 | 0.070 | 0.022 | 0.047 | 0.015 | 0.217 | 0.013 | 0.600 |
| **qwen2.5-72b** | 0.018 | 0.039 | 0.016 | 0.157 | 0.028 | 0.005 | 0.014 | 0.301 | 0.002 | 0.966 | 0.008 | 0.970 |
| **gpt-4o** | 0.011 | 0.273 | 0.024 | 0.315 | 0.041 | **0.000** | 0.013 | 0.367 | 0.030 | 0.005 | 0.021 | 0.759 |
| **Average** | 0.013 | 0.276 | 0.014 | 0.396 | 0.021 | 0.292 | 0.012 | 0.497 | 0.016 | 0.379 | 0.013 | 0.678 |

data ($\sqrt{\mathrm{CvM_r}}$). The reason we report $\sqrt{\mathrm{CvM_r}}$ is to evaluate whether deficiencies in the fit arise from a bias problem, rather than a variance problem. To compute $p$ values for (20), we use parametric bootstrapping with $B = 1000$ samples.

Table 5 indicates a close fit between the trained marginal distributions and the empirical distributions of the calibrated confidences on the test data, with an average $\sqrt{\mathrm{CvM}}$ value of 4%. However, 74% of tests reject the null hypothesis at the $p < 0.05$ level, suggesting that our model does not exactly match the data. When refitting the marginals on the test data, the average $\sqrt{\mathrm{CvM}}$ value falls to 1.5% and a much lower 18.5% of tests reject the null hypothesis. Even on the refitted data, this overall rejection rate of 18.5% is significantly higher than the 5% we would expect by chance. We conclude that our marginal distribution model fits the empirical data well, as judged by a low $\sqrt{\mathrm{CvM}}$ value, but it clearly does not capture the true distribution of calibrated confidences exactly.

Notably, the results for the refitted marginals show that the quality of the fit strongly depends on the benchmark. Specifically, TriviaQA displays a much poorer fit than the other benchmarks. For many of the LLMs, TriviaQA's low difficulty (as judged by a 90%+ test accuracy for many models) explains the poor fit. The presence of a sharp peak of calibrated confidences near $\phi_{\max}$ presumably raises the number of training samples required to precisely estimate the shape of the distribution. In addition, the ability of the beta distribution to fit sharply peaked unimodal distributions may be inherently limited. We hypothesize that these factors may explain the high $p$ values despite rather low $\sqrt{\mathrm{CvM}}$ values.

## 4.4 Rational Tuning of Confidence Thresholds

In this section, we examine the performance and runtime scaling of our continuous optimization-based algorithm (11) for selecting optimal confidence thresholds. We consider all 26 possible cascades of length $k \geq 2$ composed of Meta's Llama models (1B, 3B, 8B, 70B, and 405B). We evaluate against Bayesian optimization and high-resolution grid search baselines on six benchmarks (MMLU, MedMCQA, XSum, TriviaQA, GSM8K, TruthfulQA) spanning general-purpose knowledge and reasoning, domain-specific QA, text summarization, open-ended QA, mathematical reasoning, and the ability to avoid hallucinations on adversarial questions.

**Performance metrics**: we evaluate the area under the error-cost curve (AUC) on the test set. Specifically, computing the AUC means plotting the test error ($y$ axis) against the expected inference cost in dollars/query ($x$ axis) and evaluating the integral of this curve. Figure 5a shows an example error-cost curve and Figure 5b highlights the computation of AUC. We normalize the cost values to lie between 0 and 1, resulting in

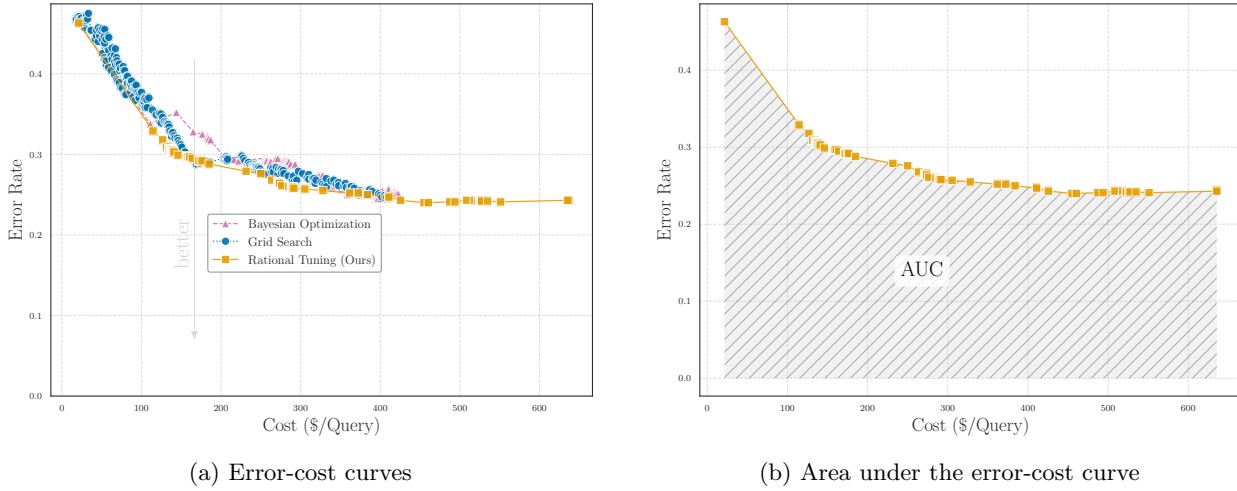

(a) Error-cost curves

(b) Area under the error-cost curve

Figure 5: Performance evaluation via the area under the error-cost curve (AUC). (a) Error-cost curves computed on the MedMCQA test set for the Llama3 3B → 8B → 70B → 405B cascade. (b) Illustration of the area under the error-cost curve (AUC).

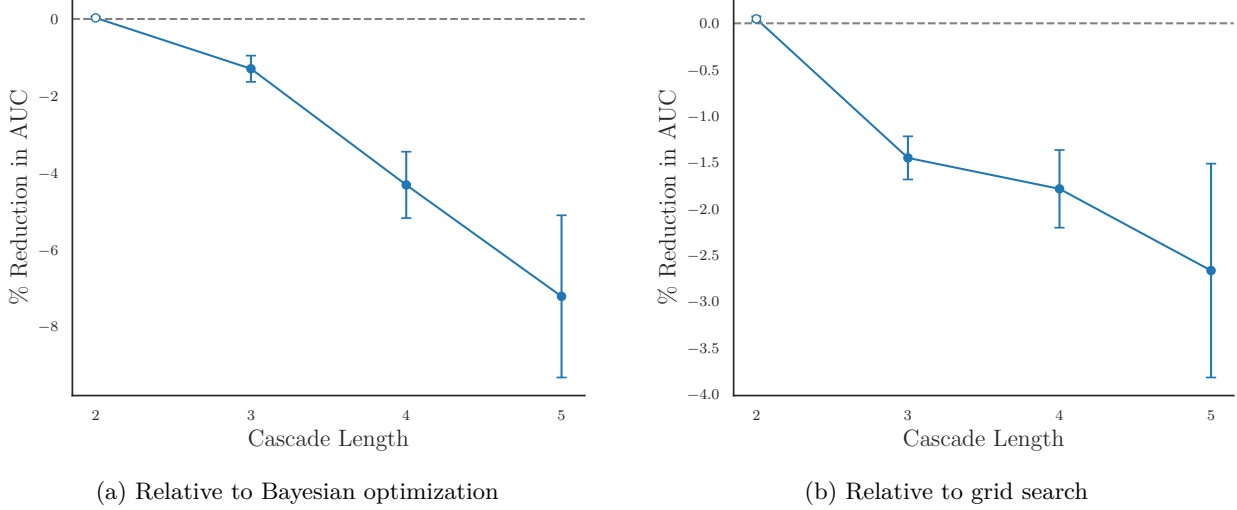

(a) Relative to Bayesian optimization

(b) Relative to grid search

Figure 6: Reduction in the area under the error-cost curve (AUC) on the test set when using our Rational Tuning framework to select confidence thresholds, as a function of cascade length. In (a), we compare against a Bayesian optimization baseline, while in (b) we compare against high-resolution grid search. For longer cascades, our method outperforms both baselines by larger margins. Error bars show $\pm 1\sigma$ of the mean percentage change, and filled markers indicate statistical significance.

AUC scores between 0 and 1 (error rate × normalized cost). Broadly, a 1% reduction in AUC means that the error rate is 1% lower at the same inference cost (on average).

In addition, we measure how the runtime for finding optimal confidence thresholds scales with the length of the cascade and the desired resolution of the error-cost curve on the $x$ axis, *i.e.*, how densely we sample the optimal thresholds. We have not overly optimized our code and mainly aim to contrast asymptotic scaling behavior.

**Bayesian optimization baseline**: this baseline runs Bayesian optimization with a Gaussian process surrogate function, via the HEBO package (Cowen-Rivers et al., 2022, Shahriari et al., 2016). The Bayesian

| Benchmark | AUC ↓ | | | RT (Ours) vs GS | | RT (Ours) vs BO | |
|---|---|---|---|---|---|---|---|
| | RT (Ours) | GS | BO | $\%\Delta_{\mathrm{GS}}$ ↓ | $p_{\mathrm{GS}}$ | $\%\Delta_{\mathrm{BO}}$ ↓ | $p_{\mathrm{BO}}$ |
| MMLU | 0.288 | **0.288** | 0.294 | 0.02 | $2.8 \times 10^{-1}$ | -2.30 | $\mathbf{4.3 \times 10^{-3}}$ |
| MedMCQA | **0.381** | 0.384 | 0.389 | -0.79 | $\mathbf{5.5 \times 10^{-3}}$ | -2.14 | $\mathbf{3.0 \times 10^{-2}}$ |
| TriviaQA | **0.181** | 0.182 | 0.183 | -0.74 | $\mathbf{6.0 \times 10^{-3}}$ | -1.58 | $\mathbf{1.0 \times 10^{-2}}$ |
| XSum | **0.409** | 0.414 | 0.416 | -1.48 | $\mathbf{2.0 \times 10^{-6}}$ | -1.92 | $\mathbf{8.9 \times 10^{-4}}$ |
| GSM8K | **0.158** | 0.162 | 0.160 | -2.68 | $\mathbf{1.0 \times 10^{-5}}$ | -1.15 | $\mathbf{1.6 \times 10^{-2}}$ |
| TruthfulQA | **0.436** | 0.437 | 0.438 | -0.26 | $\mathbf{3.8 \times 10^{-2}}$ | -0.46 | $7.9 \times 10^{-2}$ |
| Average | **0.309** | 0.311 | 0.313 | -0.99 | − | -1.60 | − |

Table 7: Area under the error-cost curve (AUC) on the test set, showing that our Rational Tuning ("RT") framework for selecting confidence thresholds consistently outperforms both a Bayesian optimization baseline ("BO") and high-resolution grid search ("GS"). The mean percentage changes ($\%\Delta$) are statistically significant at the $p < 0.05$ on almost all benchmarks, as measured by Wilcoxon rank-sum tests paired by cascade (highlighted in bold).

optimization minimizes (11), in an analogous manner to our Markov-copula ("Rational Tuning") approach. We run HEBO for as many iterations as needed until the change in loss between successive iterations is below a numerical tolerance ($\epsilon = 10^{-5}$). In practice, we found that the final change in loss is typically 0.0. Following the practical guidance of HEBO's authors[5], we use four parallel suggestions during each iteration. We adaptively interpolate the optimal thresholds computed by HEBO in the same way we do for Rational Tuning (see Equation (13)).

**High-resolution grid search baseline**: this baseline selects optimal confidence thresholds by searching over adaptive grids computed from the model-specific quantiles of calibrated confidence. Specifically, in each dimension the grid ranges from $\phi_{\min}$ to $\phi_{\max}$ in increments of 2.5% probability mass. This results in considering $40^{k-1}$ candidate threshold combinations for cascades with $k$ models, ranging from 40 candidates for a two-model cascade to $40^4 = 2{,}560{,}000$ candidates for a five-model cascade. After scoring all candidate threshold combinations, we use the Skyline operator (implemented in the Python package `paretoset`[6]) to filter the candidate threshold vectors down to the Pareto-optimal set (Börzsönyi et al., 2001). A candidate threshold vector $\boldsymbol{\theta} = (\phi_1, ..., \phi_{k-1})$ is *Pareto-optimal* if its performance metrics $(\mathbb{P}_{\boldsymbol{\theta}}(\text{Correct}), \mathbb{E}_{\boldsymbol{\theta}}[\text{Cost}])$ are not dominated by any other candidate threshold vector $\boldsymbol{\theta}'$ in the sense that $\mathbb{P}_{\boldsymbol{\theta}'}(\text{Correct}) > \mathbb{P}_{\boldsymbol{\theta}}(\text{Correct})$ and $\mathbb{E}_{\boldsymbol{\theta}'}[\text{Cost}] < \mathbb{E}_{\boldsymbol{\theta}}[\text{Cost}]$.

Figure 6 shows that our Rational Tuning framework for selecting confidence thresholds results in lower AUC on the test set compared to the baselines. Each point on the plot shows the average percentage reduction in AUC for all cascades of a given length $k$, averaged across all benchmarks. As cascade length $k$ grows, our method outperforms the baselines by a larger margin. For example, the mean reduction in AUC compared to Bayesian optimization is 4.3% for $k \geq 3$; 5.8% for $k \geq 3$; and 7.2% for $k = 5$. The corresponding performance gains relative to high-resolution grid search are 2.0%, 2.2%, and 2.7%. We computed statistical significance of these percentage differences using a Wilcoxon rank-sum test paired by cascade.

We hypothesize that the performance gains of Rational Tuning relative to Bayesian optimization stem from the fact that our framework applies a (mostly correct) inductive assumption about the correlation structure of LLM cascades, whereas Bayesian optimization is a general-purpose algorithm for black-box optimization. Section 4.4.1 corroborates this hypothesis by presenting more significant performance gains in the low-sample limit with $n \leq 30$ training examples.

By contrast, it is not surprising that grid search performs worse as $k$ increases, since searching for effective threshold combinations by trial and error suffers from the curse of dimensionality.

---

[5] `github.com/huawei-noah/HEBO/tree/master/HEBO`. Accessed April 6, 2025.
[6] Open-source implementation available at `github.com/tommyod/paretoset`. Accessed January 13, 2025.

Table 7 presents the results broken down by benchmark rather than cascade length. The table shows that Rational Tuning consistently outperforms Bayesian optimization and grid search across benchmarks, independent of cascade length. The only benchmark mark where we report a tie is MMLU. On almost all benchmarks, the reductions in AUC are statistically significant at the $p < 0.05$ level.

### 4.4.1 Performance in the Low-Sample Limit

Our Rational Tuning methodology relies on a small labeled training data set consisting of LLM confidence scores and corresponding binary correctness labels. Since labeled data is scarce in many applications, we supplement our main experiment ($n \approx 300$ training examples) with a study of the low-sample limit. Here, we re-run our experiment for $n \leq 30$ training examples. For each benchmark, we ensure a balanced subsample with both correct and incorrect answers from each model by sampling training examples in pairs (one correct, one incorrect for each model) for a fixed number of iterations, with a target of $n = 30$ examples. Since collisions may occur, the final number of sampled training examples lies between 20 and 30, depending on the benchmark.

Figure 7 displays the results, revealing that Rational Tuning significantly outperforms the Bayesian optimization and grid search baselines as cascade length grows. On cascades with $k \geq 3$ models, the average performance gain is 10.2% relative to Bayesian optimization, and 5.6% relative to high-resolution grid search.

Table 8 breaks down these results by benchmark. We see that Rational Tuning outperforms the baselines on each benchmark except for TruthfulQA, where high-resolution grid search performs best. On the other benchmarks (MMLU, MedMCQA, TriviaQA, XSum, and GSM8K), the performance gains of our method are statistically significant at the $p < 0.05$ level, according to Wilcoxon rank-sum tests paired by cascade.

Our interpretation of these results is that our Rational Tuning framework benefits from making inductive assumptions about the interactions between the error rates of different LLMs. Crucially, fitting these inductive assumptions to empirical data requires only few observations: since each copula model depends on a single scalar correlation parameter $\theta \in \mathbb{R}$, our method requires only $k - 1$ parameters to model the interactions between the error rates of $k$ different LLMs.

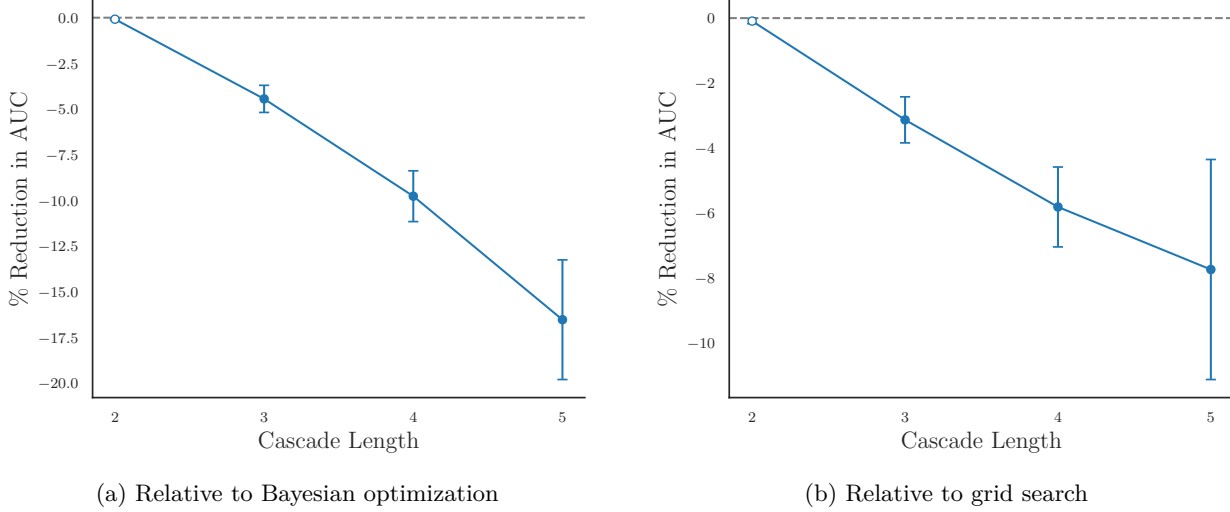

(a) Relative to Bayesian optimization

(b) Relative to grid search

Figure 7: Reduction in the area under the error-cost curve (AUC) as cascade length grows, in the low-sample limit ($n \leq 30$ training examples), when using our Rational Tuning framework. In (a), we compare against a Bayesian optimization baseline, while in (b) we compare against high-resolution grid search. Our method increasingly outperforms the baselines as cascade length grows. Error bars show $\pm 1\sigma$ of the mean percentage change, and filled markers indicate statistical significance.

| Benchmark | AUC ↓ | | | RT (Ours) vs GS | | RT (Ours) vs BO | |
|---|---|---|---|---|---|---|---|
| | RT (Ours) | GS | BO | %$\Delta_{\text{GS}}$ ↓ | $p_{\text{GS}}$ | %$\Delta_{\text{BO}}$ ↓ | $p_{\text{BO}}$ |
| MMLU | **0.293** | 0.308 | 0.316 | -5.51 | $\mathbf{8.0 \times 10^{-6}}$ | -7.74 | $\mathbf{1.0 \times 10^{-6}}$ |
| MedMCQA | **0.399** | 0.416 | 0.419 | -4.18 | $\mathbf{1.1 \times 10^{-5}}$ | -4.72 | $\mathbf{5.0 \times 10^{-6}}$ |
| TriviaQA | **0.197** | 0.200 | 0.203 | -2.11 | $\mathbf{1.5 \times 10^{-2}}$ | -3.26 | $\mathbf{5.1 \times 10^{-3}}$ |
| XSum | **0.422** | 0.431 | 0.442 | -2.42 | $\mathbf{2.6 \times 10^{-2}}$ | -5.06 | $\mathbf{4.2 \times 10^{-4}}$ |
| GSM8K | **0.187** | 0.195 | 0.197 | -3.41 | $\mathbf{2.5 \times 10^{-2}}$ | -4.21 | $\mathbf{2.1 \times 10^{-4}}$ |
| TruthfulQA | 0.449 | **0.442** | 0.452 | 1.60 | $1.0 \times 10^{0}$ | -0.73 | $1.2 \times 10^{-1}$ |
| Average | **0.325** | 0.332 | 0.338 | -2.67 | – | -4.29 | – |

Table 8: Area under the error-cost curve (AUC) in the low-sample limit ($n \leq 30$ training examples), showing that our Rational Tuning ("RT") framework for selecting confidence thresholds consistently outperforms both a Bayesian optimization baseline ("BO") and high-resolution grid search ("GS"). The mean percentage changes (%$\Delta$) are statistically significant at the $p < 0.05$ on almost all benchmarks, as measured by Wilcoxon rank-sum tests paired by cascade (highlighted in bold).

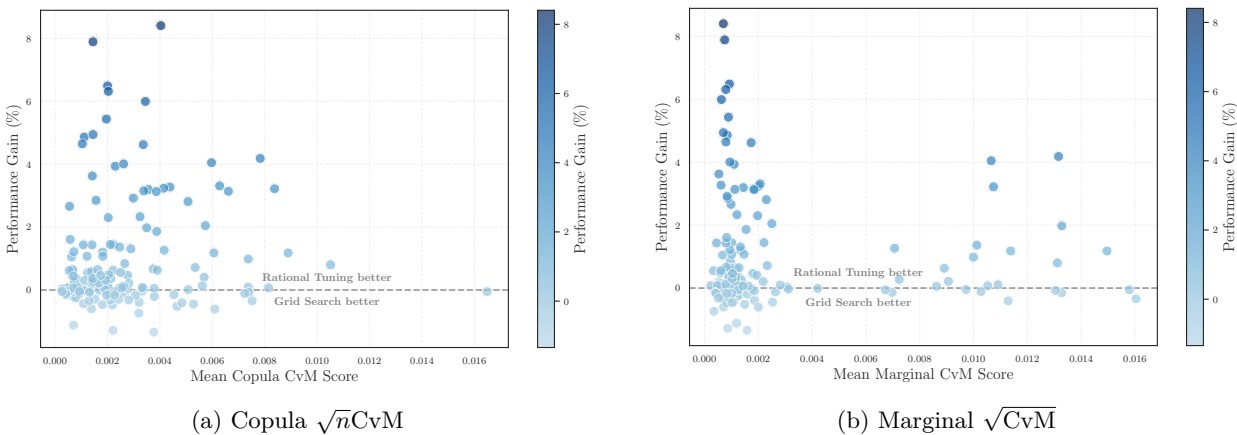

(a) Copula $\sqrt{n}$CvM

(b) Marginal $\sqrt{\text{CvM}}$

Figure 8: Sensitivity of Rational Tuning's performance gains to the Cramér-von Mises (CvM) test statistics (lower is better). Overall, performance appears to be more sensitive to mis-specification of the copula model.

### 4.4.2 Sensitivity to Statistical Assumptions

Our implementation of Rational Tuning models uses mixtures of beta distributions to model the marginal distribution of confidence scores, and Gumbel copulas to model pairwise correlations between LLMs. In Section 4.3, we quantify the deviation between these modeling assumptions and the true empirical distributions via Cramér-von Mises (CvM) statistics.

Figure 8 shows the sensitivity of Rational Tuning's performance gains (relative to the high-resolution grid search baseline) to the mean CvM statistics for each cascade. For example, if $M_1 \to ... \to M_k$ has marginal $\sqrt{\text{CvM}}$ scores $\sigma_1, ..., \sigma_k$ and copula $\sqrt{n}$CvM scores $\sigma_{1,2}, \sigma_{2,3}, ..., \sigma_{k-1,k}$, then the mean marginal and copula CvM scores are $\overline{\sigma}_{\text{marginal}} = \frac{1}{k} \sum_{i=1}^{k} \sigma_i$, and $\overline{\sigma}_{\text{copula}} = \frac{1}{k} \sum_{i=2}^{k} \sigma_{i-1,i}$, respectively.

Figures 8a and 8b suggest that lower CvM divergences improve the relative performance of Rational Tuning. This effect is more pronounced for the copula statistics rather than the marginal statistics, highlighting the importance of correctly modeling the correlations between LLMs. In both plots, the light blue data points with little performance gain despite excellent CvM values are heavily enriched for two-model cascades. For cascades with $k = 2$ models, Rational Tuning generally performs on par with grid search or Bayesian optimization.

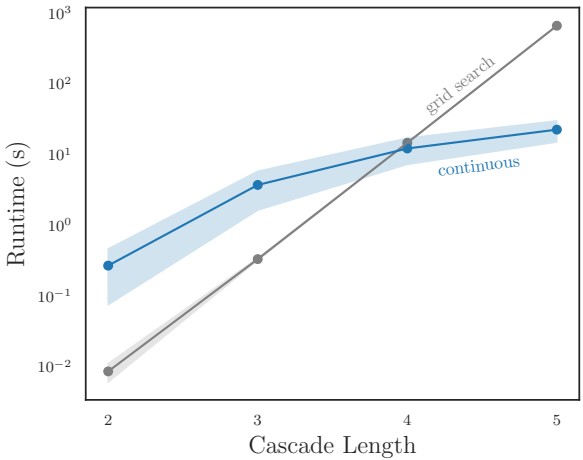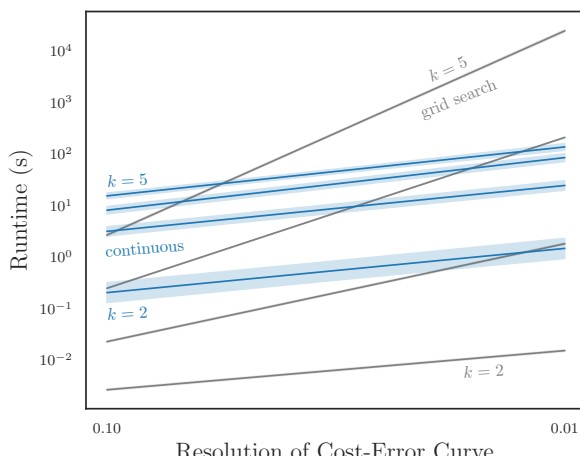

(a) The runtime of grid search grows exponentially in the cascade length, whereas the runtime of our method grows as a low-order polynomial (semilog-y plot).

(b) The runtime of our method always grows linearly in the desired resolution $h$ at which the error-cost curve is sampled along the cost axis, independent of the cascade length $k$. However, the runtime of grid search scales as $h^{k-1}$ in $h$ and $k$ (log-log plot).

Figure 9: Shows runtime scaling for computing the full error-cost curve, comparing our continuous-optimization based algorithm ("continuous", blue) to grid search ("grid search", gray). Our method scales much more favorably as the cascade length grows, and as the error-cost curve is sampled more densely along the cost axis. The shading shows $\pm 1\sigma$ of the observed data points.

The plots also suggest some robustness to deviations from the statistical assumptions. We are eager to explore Rational Tuning's robustness to model mis-specification in greater detail in subsequent work.

### 4.4.3 Computational Scaling

Moving on to a comparison of computational complexity, Figure 9 shows that the runtime of our Rational Tuning framework for finding optimal confidence thresholds scales much more favorably compared to grid search, both in the length of the cascade $k$ as well as the desired resolution $h$ of the error-cost curve. Here, the resolution $h$ refers to the density at which we sample the optimal error-cost curve along the cost-axis. For grid search, $h$ is simply the reciprocal of the number of grid points in each dimension. For our method, $h$ is the reciprocal of the number of times we solve the optimization problem (11). In other words, $h = 1/|\Lambda|$, where $\Lambda$ is the set of cost sensitivities we consider in (11).

We omit Bayesian optimization from Figure 9, since we observed longer runtimes (10-1000x longer) that exhibit less clear scaling with the length $k$ of the cascade. Specifically, the average number of iterations until convergence increases from 2.9 for $k = 2$ to 5.3 for $k = 4$, then drops back to 4.0 for $k = 5$. Across the data, the minimum and maximum number of iterations required until convergence are 2 and 14.

### 4.4.4 Practical Guidelines

Our experiments suggest that Rational Tuning is the preferred methodology for tuning confidence thresholds for longer cascades ($k \geq 3$) with multiple deferral thresholds. However, for two-model cascades ($k = 2$) with a single deferral threshold, the Markov assumption is void (as there are only two models). In this setting, the nonparametric nature of one-dimensional grid search should give the most reliable results.

When applying Rational Tuning for cascades with $k \geq 3$ models, we recommend visually inspecting the match between the assumed probabilistic model and the empirical data. Specifically, we recommend the following visual diagnostics:

- **Marginal distributions**: compare histograms of the fitted and empirical distributions (as in Figure 4) and verify that the overall fit is adequate.

- **Pairwise correlations**: construct copula plots as in Figures 2 and 3. Compare these plots to random samples from a Gumbel copula with correlation parameter $\hat{\theta} = \frac{1}{1-\hat{\tau}}$, where $\hat{\tau}$ is the empirical rank correlation.

- **Markov assumption**: construct rank correlation plots as in Figure 1. Verify that rank correlations are strong near the diagonal.

In addition, it is important to assess the expected calibration error (ECE) of the confidence scores. We recommend computing the ECE using quantile binning with 10 or 20 bins; ideally, the ECE should not exceed 10%.

## 5 Conclusion

We have presented a framework for rationally tuning the confidence thresholds of LLM cascades using continuous optimization. Our approach is based on a parametric probabilistic model for the calibrated confidences of a sequence of LLMs. This probabilistic model is based on a Markov factorization, which accounts for pairwise correlations between the error rates of different LLMs using copulas, yielding a data-efficient approach. Goodness-of-fit analyses spanning 10 LLMs and 6 benchmarks have shown good agreement with the test data.

Importantly, our probabilistic model yields analytical expressions for a cascade's error rate and expected inference cost. These expressions are differentiable with respect to the cascade's confidence thresholds, making continuous optimization possible. Compared to selecting confidence thresholds using Bayesian optimization and high-resolution grid search, our Rational Tuning framework yields more favorable error-cost trade-offs as cascade length grows, outperforming the baselines by up to 7.2% when using $n \approx 300$ labeled training examples. In the low-sample limit ($n \leq 30$ training examples), the performance gains reach up to 16.5%, suggesting that our framework's inductive assumptions about the interactions between the error rates of different LLMs improve sample efficiency.

Building on these promising results, an interesting direction would be to apply our probabilistic modeling framework to LLM routing, in which a central routing model sends a query to the most suitable LLM in a single step, avoiding cumulative cost increases as the query propagates down a cascade. Since cumulative cost increases are especially severe for longer cascades (at which our methodology excels), the routing setting may more effectively leverage Rational Tuning's capacity for modeling dependencies between arbitrarily many distinct LLMs. For instance, suppose the routing decision depends on noisy estimates $\hat{\phi}_1, ..., \hat{\phi}_n$ of the LLMs' true calibrated confidences. In this case, balancing the noisy observations $\hat{\phi}_i$ against their probabilistic expectations $\mathbb{E}[\hat{\phi}_i | \hat{\phi}_1, ..., \hat{\phi}_{i-1}, \hat{\phi}_{i+1}, ... \hat{\phi}_n]$ may lead to more effective routing decisions.

Ultimately, our results point to a larger vision for the future of deploying LLMs. Using probabilistic models, we will be able to adaptively select the most suitable model to answer each query, improving both reliability and performance. Additionally, probabilistic modeling will enable us to anticipate the performance of a system of LLMs under different conditions, making it possible to seamlessly adapt the system as conditions shift. We are excited to further pursue this line of research in subsequent work.

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

## Appendix A: Proof of Proposition 2

**Proposition 2.** *Consider a cascade $M_1 \to ... \to M_k$ with confidence thresholds $(\phi_1, ..., \phi_{k-1})$. Assume that the distribution functions for the calibrated confidences $\Phi_i$ satisfy (5), for $i = 1, 2, ..., k$. Assume further that the expected numbers of input and output tokens, $T_i^{(in)}$ and $T_i^{(out)}$, for each model $i$ are independent of the calibrated confidences $\Phi_1, ..., \Phi_k$. Then the probability of correctness $\mathbb{P}(Correct)$ and expected cost $\mathbb{E}[Cost]$ for the cascade are*

$$\mathbb{P}(Correct) = \int_{\{\Phi_1 > \phi_1\}} \Phi_1(\omega) \ d\mathbb{P}(\omega) \tag{21}$$
$$+ \sum_{i=2}^{k} \mathbb{P}(\Phi_1 \le \phi_1) \left( \prod_{j=2}^{i-1} \mathbb{P}(\Phi_j \le \phi_j | \Phi_{j-1} \le \phi_{j-1}) \right) \int_{\{\Phi_i > \phi_i\}} \Phi_i(\omega) \ d\mathbb{P}(\omega | \Phi_{i-1} \le \phi_{i-1})$$
$$\mathbb{E}[Cost] = (1 - \mathbb{P}(\Phi_1 \le \phi_1)) \ \mathbb{E}[C_1] \tag{22}$$
$$+ \sum_{i=2}^{k} \mathbb{P}(\Phi_1 \le \phi_1) \left( \prod_{j=2}^{i-1} \mathbb{P}(\Phi_j \le \phi_j | \Phi_{j-1} \le \phi_{j-1}) \right) (1 - \mathbb{P}(\Phi_i \le \phi_i | \Phi_{i-1} \le \phi_{i-1})) \sum_{j=1}^{i} \mathbb{E}[C_j],$$

*where $C_i$ is the cost per query of model $i$. Specifically, if $\gamma_i^{(in)}$ and $\gamma_i^{(out)}$ are the costs per input and output token, $C_i = \gamma_i^{(in)} T_i^{(in)} + \gamma_i^{(out)} T_i^{(out)}$. To simplify the notation, we let $\phi_k := -\infty$ (although there is no confidence threshold for the final model in the cascade).*

*Proof.* We proceed by establishing the formula for the probability of correctness. Analogous reasoning then yields the formula for expected cost. Let $\tau \in \{1, ..., k\}$ be the index of the model $M_\tau$ that returns the query. Specifically, $\{\tau = i\} = \{\Phi_1 \le \phi_1, ..., \Phi_{i-1} \le \phi_{i-1}, \Phi_i > \phi_i\}$. We will decompose $\mathbb{P}(\text{Correct})$ based on the value of $\tau$. First, since the calibrated confidence $\Phi_i$ satisfies $\Phi_i = \mathbb{E}[\mathbb{1}\{M_i \text{ correct}\}|x]$, we have

$$\mathbb{P}(\text{correct}) = \mathbb{E}[\Phi_\tau] = \mathbb{E}\Big[ \sum_{i=1}^{k} \Phi_i \mathbb{1}\{\tau = i\} \Big] = \sum_{i=1}^{k} \mathbb{E}[\Phi_i \mathbb{1}\{\tau = i\}]. \tag{23}$$

Hence, the problem reduces to computing $\mathbb{E}[\Phi_i \mathbb{1}\{\tau = i\}]$ for each model $i$. This is the integral of $\Phi_i$ over the set $\{\tau = i\}$. For $i \ge 2$, we have

$$\mathbb{E}[\Phi_i \mathbb{1}\{\tau = i\}] = \int \Phi_i \mathbb{1}_{\{\Phi_i > \phi_i\}} \prod_{j=1}^{i-1} \mathbb{1}_{\{\Phi_j \le \phi_j\}} \ d\mathbb{P} \tag{24}$$

$$= \mathbb{P}(\Phi_1 \le \phi_1, ..., \Phi_{i-1} \le \phi_{i-1}) \int \Phi_i \mathbb{1}_{\{\Phi_i > \phi_i\}} \frac{\prod_{j=1}^{i-1} \mathbb{1}_{\{\Phi_j \le \phi_j\}}}{\mathbb{P}(\Phi_1 \le \phi_1, ..., \Phi_{i-1} \le \phi_{i-1})} \ d\mathbb{P} \tag{25}$$

$$= \mathbb{P}(\Phi_1 \le \phi_1) \prod_{j=2}^{i-1} \mathbb{P}(\Phi_j \le \phi_j | \Phi_{j-1} \le \phi_{j-1}) \int_{\{\Phi_i > \phi_i\}} \Phi_i \ d\mathbb{P}(\cdot | \cap_{j=1}^{i-1} \{\Phi_j \le \phi_j\}) \tag{26}$$

$$= \mathbb{P}(\Phi_1 \le \phi_1) \prod_{j=2}^{i-1} \mathbb{P}(\Phi_j \le \phi_j | \Phi_{j-1} \le \phi_{j-1}) \int_{\{\Phi_i > \phi_i\}} \Phi_i \ d\mathbb{P}(\cdot | \Phi_{i-1} \le \phi_{i-1}). \tag{27}$$

To obtain Equation (26), we applied the Markov assumption (5) and switched from the standard probability measure $\mathbb{P}(\cdot)$ to the conditional probability measure $\mathbb{P}(\cdot \cap A)/\mathbb{P}(A)$, where $A = \{\Phi_1 \le \phi_1, ..., \Phi_{i-1} \le \phi_{i-1}\}$. To obtain the last line, we applied the Markov assumption (5) again.

For $i = 1$, we have that

$$\mathbb{E}[\Phi_1 \mathbb{1}\{\tau = 1\}] = \int_{\{\Phi_1 > \phi_1\}} \Phi_1(\omega) \ d\mathbb{P}(\omega). \tag{28}$$

This concludes the proof of the formula for the probability of correctness. To obtain the formula for the expected cost, we reason analogously and note that the integral

$$\int_{\Phi_i > \phi_i} \sum_{j=1}^{i} C_j \ \mathrm{d}\mathbb{P}(\cdot|\Phi_{i-1} \le \phi_{i-1}) \tag{29}$$

simplifies to the product $\mathbb{P}(\Phi_i > \phi_i|\Phi_{i-1} \le \phi_{i-1}) \sum_{j=1}^{i} \mathbb{E}[C_j]$ because we assume the model costs to be independent of the calibrated confidences. $\qquad\square$

## Appendix B: Algorithm for Computing $\mathbb{P}(\textbf{Correct})$ and $\mathbb{E}[\textbf{Cost}]$

---

**Algorithm 1** Computing $\mathbb{P}(\text{Correct})$ and $\mathbb{E}[\text{Cost}]$

---

**Require:** confidence thresholds $\phi_1, ..., \phi_{k-1} \in \mathbb{R}^{k-1}$

1: `cum_cost` $\leftarrow \mathbb{E}[C_1]$            *# cumulative expected cost*
2: `cum_transition_prob` $\leftarrow 1$        *# cumulative transition probability*
3: `correctness_terms` $\leftarrow [\ ]$       *# expected correctness due to different models*
4: `cost_terms` $\leftarrow [\ ]$           *# expected costs due to different models*
5: $\phi_k \leftarrow -\infty$
6:
7: `correctness_terms`.append($\int_{\{\Phi_1 > \phi_1\}} \Phi_1(\omega) \ \mathrm{d}\mathbb{P}(\omega)$)
8: `cost_terms`.append($(1 - \mathbb{P}(\Phi_1 \le \phi_1)) \times$ `cum_cost`)
9: `cum_transition_prob` $\leftarrow$ `cum_transition_prob` $\times \mathbb{P}(\Phi_1 \le \phi_1)$
10:
11: **for** $i = 2 \ ... \ k$ **do**
12:     `cum_cost` $\leftarrow$ `cum_cost` $+ \mathbb{E}[C_i]$
13:     `correctness_terms`.append(`cum_transition_prob` $\times \int_{\{\Phi_i > \phi_i\}} \Phi_i(\omega) \ \mathrm{d}\mathbb{P}(\omega|\Phi_{i-1} \le \phi_{i-1})$)
14:     `cost_terms`.append(`cum_transition_prob` $\times (1 - \mathbb{P}(\Phi_i \le \phi_i|\Phi_{i-1} \le \phi_{i-1})) \times$ `cum_cost`)
15:     `cum_transition_prob` $\leftarrow$ `cum_transition_prob` $\times \mathbb{P}(\Phi_i \le \phi_i|\Phi_{i-1} \le \phi_{i-1})$
16: **end for**
17:
18: $\mathbb{P}(\text{Correct}) \leftarrow \text{sum}(\texttt{correctness\_terms})$
19: $\mathbb{E}[\text{Cost}] \leftarrow \text{sum}(\texttt{cost\_terms})$
20:
21: **return** $(\mathbb{P}(\text{Correct}), \mathbb{E}[\text{Cost}])$

---

Algorithm 1 provides an efficient way to compute the probability of correctness and expect cost in $\mathrm{O}(k)$ time, where $k$ is the length of the cascade. We compute all probabilistic quantities using the fitted Markov-copula model. To compute the integrals

$$I_i(\phi_{i-1}, \phi_i) = \int_{\{\Phi_i > \phi_i\}} \Phi_i(\omega) \ \mathrm{d}\mathbb{P}(\omega|\Phi_{i-1} \le \phi_{i-1}) \tag{30}$$

of conditional correctness, we use numerical integration by treating (30) as a Riemann-Stieltjes integral $\int_{\phi_1}^{1} \phi \ \mathrm{d}F(\phi)$ in the distribution function $F(\phi) = \mathbb{P}(\Phi_i \le \phi|\Phi_{i-1} \le \phi_{i-1})$. See Rudin (1976). Before solving the minimization problem (11), we pre-compute look-up tables for $I_i(\phi_{i-1}, \phi_i)$ which can be re-used when solving (11) for different values of $\lambda$ and different subcascades.

## Appendix C: Prompt Templates

Below, we provide the exact text of the prompts used in our experiments. Placeholders (for example, `{question}`) are replaced at runtime with the relevant content.

## 5.1  MMLU

**User Prompt (Zero-Shot)**

```
Answer the multiple-choice question below by outputting A, B, C, or D.
Don't say anything else.

Question: {question}

Choices:
{choices}

Answer:
```

**System Prompt**

```
Correctly answer the given multiple-choice question by outputting "A", "B",
"C", or "D". Output only "A", "B", "C", or "D", nothing else.
```

## 5.2  MedMCQA

**User Prompt (Zero-Shot)**

```
Below is a multiple-choice question from a medical school entrance exam.
Output "A", "B", "C", or "D" to indicate the correct answer.
Don't say anything else.

Question: {question}

Choices:
{choices}

Answer:
```

**System Prompt**

```
Your job is to answer a multiple-choice question from a medical school
entrance exam. Correctly answer the question by outputting "A", "B", "C",
or "D". Output only "A", "B", "C", or "D", nothing else.
```

## 5.3  TriviaQA

**User Prompt (Zero-Shot)**

```
Correctly answer the question below. Give the answer directly,
without writing a complete sentence.

Question: {question}

Answer:
```

**System Prompt**

```
Correctly answer the given question. Answer the question directly
without writing a complete sentence. Output just the answer, nothing else.
```

**Evaluation User Prompt**

```
Consider a proposed answer to the following trivia question: {question}.
The proposed answer is {model_answer}. Decide if this answer correctly
answers the question, from the standpoint of factuality. Output "Y" if
the answer is factually correct, and "N" otherwise. Do not say anything else.
```

**Evaluation System Prompt**

```
You are a helpful assistant who judges answers to trivia questions. Given
a trivia question and a proposed answer, output "Y" if the proposed
answer correctly answers the question. Otherwise, if the answer is not
factually correct, output "N". Only output "Y" or "N". Do not say anything else.
```

## 5.4 XSum

**User Prompt (Zero-Shot)**

```
Summarize the given source document. Write a concise summary that is coherent,
consistent, fluent, and relevant, as judged by the following criteria:

Coherence - collective quality of all sentences
Consistency - factual alignment between the summary and the source
Fluency - quality of individual sentences
Relevance - selection of important content from the source

Source document: {source_document}

Summary:
```

**System Prompt**

```
Summarize the given document. Output only the summary, and nothing else.
Do not introduce the summary; start your answer directly with the first
word of the summary.
```

**Evaluation User Prompt**

```
Consider a proposed summary of the following source document: {source_document}.
Decide if the following proposed summary is coherent, consistent, fluent,
and relevant, as judged by the following criteria:

Coherence - collective quality of all sentences
Consistency - factual alignment between the summary and the source
Fluency - quality of individual sentences
Relevance - selection of important content from the source

Score each criterion (coherence, consistency, fluency, and relevance)
on a scale from 1-5, where 5 is best. Return a JSON of the form
{"coherence": a, "consistency": b, "fluency": c, "relevance": d},
where a, b, c, d are the scores for the criteria (1-5). Only return this JSON.

Proposed summary: {model_answer}

JSON containing the scores for all criteria:
```

**Evaluation System Prompt**

```
You are a helpful assistant who evaluates the quality of text summaries
based on coherence, consistency, fluency, and relevance, as judged by the
following criteria:

Coherence - collective quality of all sentences
Consistency - factual alignment between the summary and the source
Fluency - quality of individual sentences
Relevance - selection of important content from the source

Score each criterion on a scale from 1-5 (5 is best). Only respond with
a JSON. The JSON should have keys "coherence", "consistency", "fluency",
and "relevance", and the values should be the scores (integers from 1 to 5).
```

## 5.5 GSM8K

**User Prompt (Zero-Shot)**

```
Reason step-by-step through the following math problem. End your answer
by outputting the final numerical answer prefixed by '#### '. For example,
if your final answer is x, end your answer on '#### x'.

Problem: {question}

Step-By-Step Reasoning:
```

**System Prompt**

```
Consider the given math problem, think step-by-step, then report the final
numerical answer x as follows '#### x'. Do not say anything after that.
```

**Evaluation User Prompt**

```
Consider a proposed answer to the following math problem: {question}.
Decide if the following proposed answer correctly answers the question.
Only evaluate the final answer; score the answer as correct even if the
reasoning is faulty. For reference, the correct answer is provided below.

Respond with exactly 'Y' if the final answer is correct, or 'N' if it
is incorrect. Only output Y or N.

Proposed answer: {model_answer}

Correct answer:
{correct_answer}

Is the given final answer correct? Respond with exactly Y or N:
```

**Evaluation System Prompt**

```
You are a helpful assistant who evaluates answers. Only respond with Y or N.
```

## 5.6 TruthfulQA

**User Prompt (Zero-Shot)**

```
Answer the question below truthfully and concisely. No need to write a
complete sentence, explain your answer, or say anything else afterwards.

Question: {question}

Answer:
```

**System Prompt**

```
Your task is to answer the given question truthfully. It is okay to
acknowledge you do not know the answer. Give the correct answer if you can.
```

**Evaluation User Prompt**

```
Consider a proposed answer to the following question: {question}. Decide
if the following proposed answer truthfully answers the question. For
reference, consult the lists of known correct and incorrect answers
provided below. Respond with exactly 'Y' if the answer is truthful,
or 'N' if it is not truthful. Only output Y or N.

Proposed answer: {model_answer}

Known correct answers:
{make_bullets(correct_answers)}

Known incorrect answers:
{make_bullets(incorrect_answers)}

Is the given answer truthful? Respond with exactly Y or N:
```

**Evaluation System Prompt**

```
You are a helpful assistant who evaluates answers. Only respond with Y or N.
```

## Appendix D: Price Differentials between Small and Large Models

Table 9 lists the differentials between smaller and larger language models across various providers.

Table 9: Price differentials between smaller and larger language models across various providers. Ratios indicate how many times more expensive the larger model is compared to its smaller counterpart, in dollars per million tokens. Data as of December 20th, 2024.

| Δ Intelligence | Provider | Smaller Model | Larger Model | Price Ratio |
|---|---|---|---|---|
| **Small Gap** | Meta | llama3.1-70b | llama3.1-405B | 3.33x |
| | Anthropic | claude-3.5-sonnet | claude-3-opus | 5.00x |
| | OpenAI | gpt4o | o1 | 6.00x |
| **Medium Gap** | Meta | llama3.1-8b | llama3.1-405b | 15.0x |
| | OpenAI | gpt4o-mini | gpt4o | 16.67x |
| | Anthropic | claude-3.5-haiku | claude-3-opus | 18.75x |
| **Large Gap** | Meta | llama3.2-1b | llama3.1-405b | 30.0x |
| | Anthropic | claude-3-haiku | claude-3-opus | 60.0x |
| | OpenAI | gpt4o-mini | o1 | 100.0x |

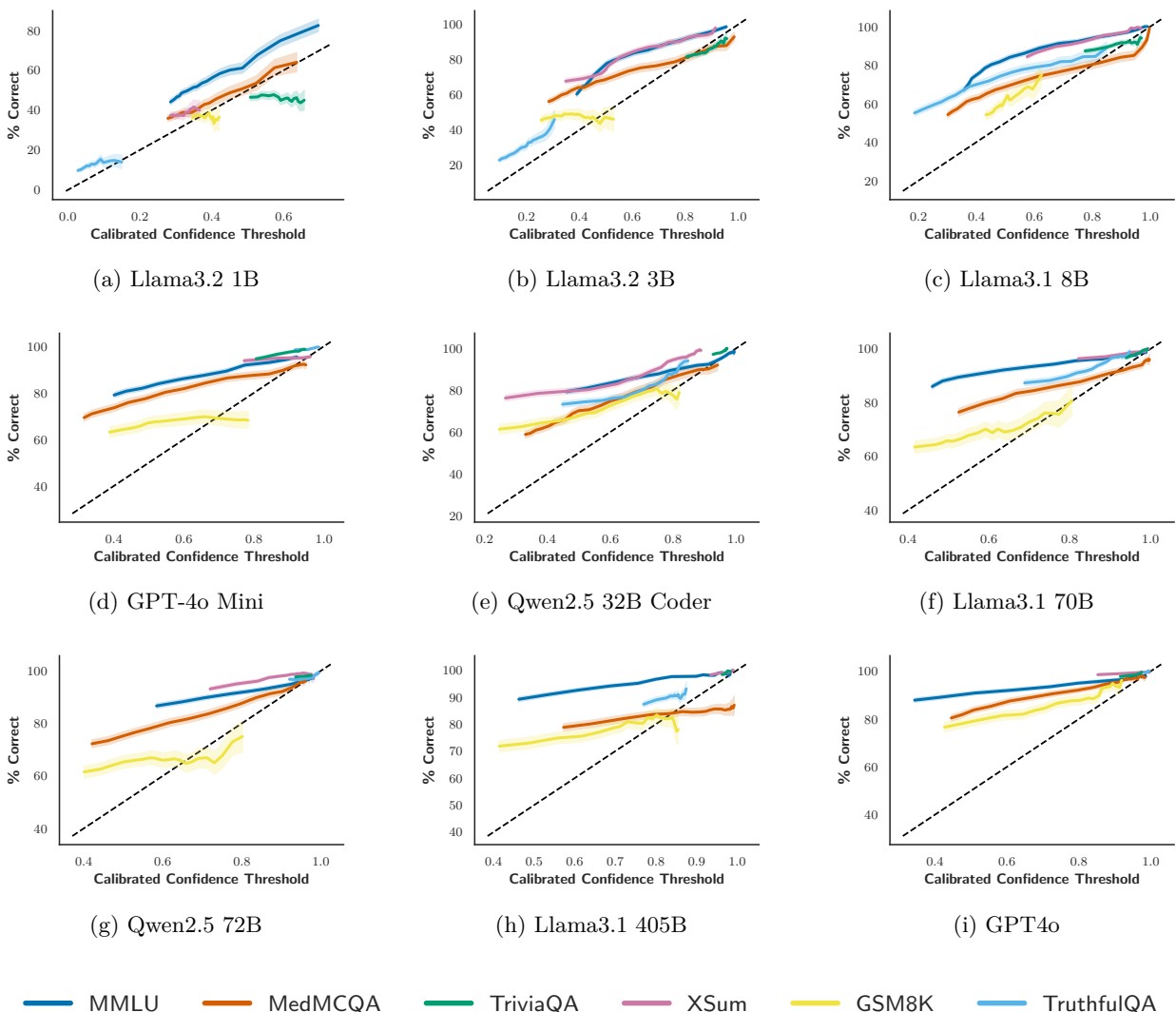

Figure 10: Verifies that confidence thresholding works by showing that for most benchmarks and models, test accuracy increases to above $q$ when only accepting queries on which the calibrated confidence for the query exceeds $q$. Calibration was performed on the training set. The shading indicates $\pm 1\sigma$, as computed by a binomial model for the number of correct answers. Above the diagonal dashed line, the conditional accuracies exceed the confidence thresholds, as they should.

## Appendix E: Verifying Confidence Thresholding on the Test Sets

We further verify calibration of LLM confidences by showing that confidence thresholding works: for most benchmarks and models, when only accepting queries for which the calibrated confidence exceeds $q$, the test error decreases to below $< 1 - q$.

Figure 10 plots the conditional accuracy with confidence thresholding on the test sets ($n \approx 1000$). In each case, the logistic regression calibrator was fitted on the training set ($n \approx 300$). Each plot traces the empirical probability of correctness on the test set, $\hat{\mathbb{P}}_{\text{test}}(\text{correct}|\Phi \geq \phi)$, for different values of the calibrated confidence threshold $\phi$. The figure shows that, for the most part, the models' conditional accuracies increase as expected. This is indicated by the fact that the conditional accuracy curves mostly remain above the diagonal dashed lines, reflecting the theoretical expectation that $\hat{\mathbb{P}}_{\text{test}}(\text{correct}|\Phi \geq \phi) \geq \phi$.

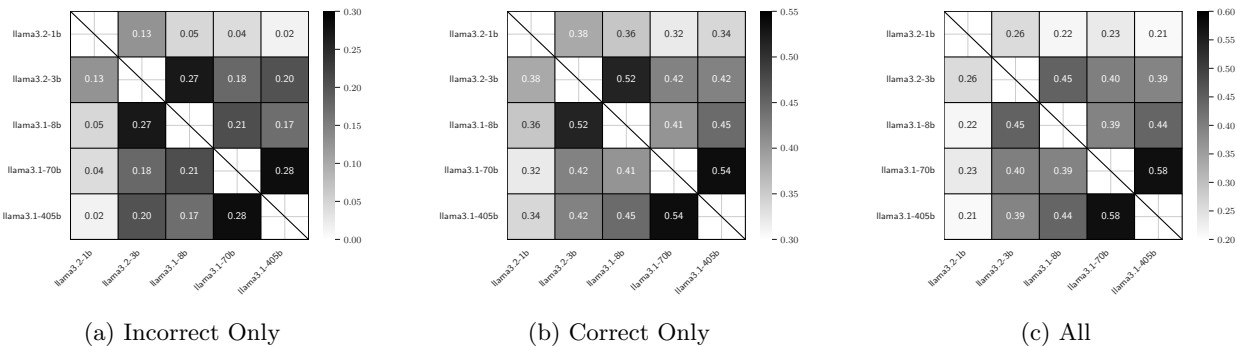

(a) Incorrect Only           (b) Correct Only           (c) All

Figure 11: MMLU: Kendall's $\tau$ rank correlations of Llama3 models ordered by size.

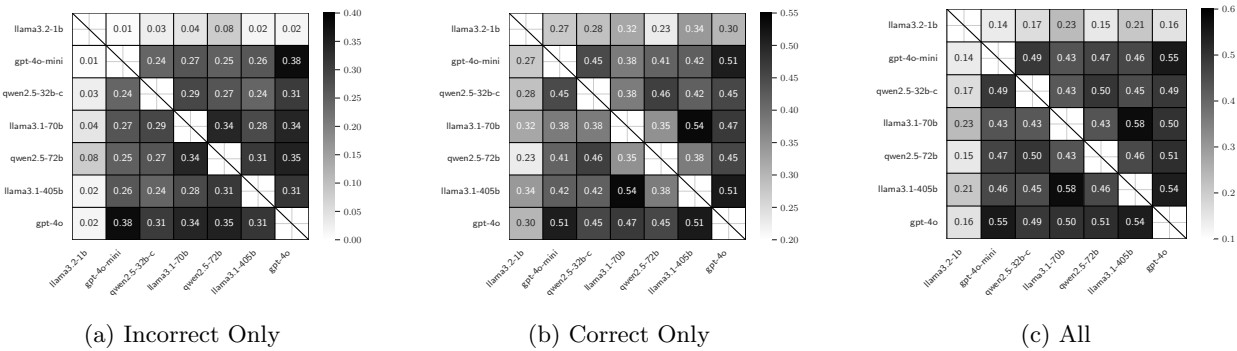

(a) Incorrect Only           (b) Correct Only           (c) All

Figure 12: MMLU: Kendall's $\tau$ rank correlations of Llama3, GPT-4o, and Qwen2.5 models ordered by size.

## 6    Appendix F: Recomputing Rank Correlations on Correct and Incorrect Answers

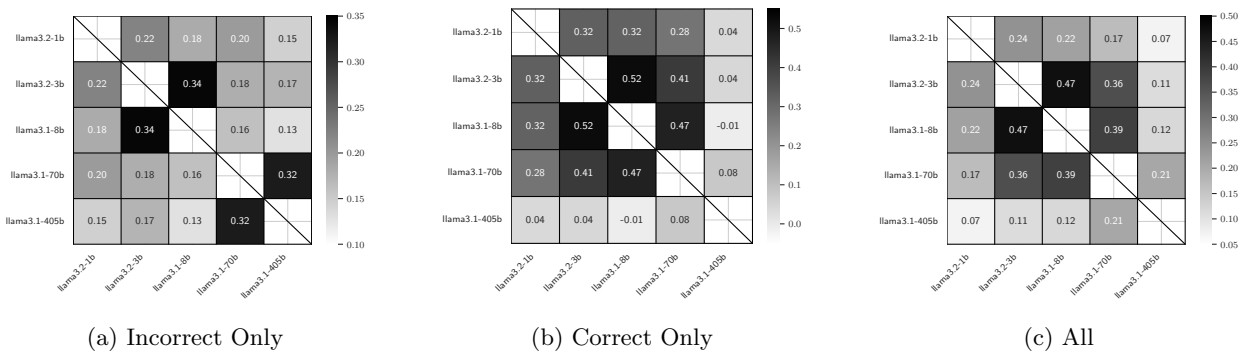

(a) Incorrect Only           (b) Correct Only           (c) All

Figure 13: MedMCQA: Kendall's $\tau$ rank correlations of Llama3 models ordered by size.

In this section, we verify the rank correlations between the confidence scores of different LLMs by recomputing them conditioned on both models answering correctly or incorrectly.

Figures 11-22 extend Figure 1 by separately re-computing the rank correlation patterns on correctly and incorrectly answered queries. In addition, Table 10 below shows the average rank correlations computed separately on the *correct*, *incorrect*, and *all answers*, for each benchmark. We compute $\tau_{\text{inc}}$, $\tau_{\text{corr}}$, $\tau_{\text{all}}$ for each pair of models, as well as the rank correlations between these measurements across model pairs: $\tau_{\text{inc, corr}}$, $\tau_{\text{inc, all}}$, $\tau_{\text{corr, all}}$.

Note that since error rates are low for some models and benchmarks (see Table 1), conditioning on incorrectly answered queries leaves only few observations for some model pairs. In Figures 11-22, we print "?" for rank

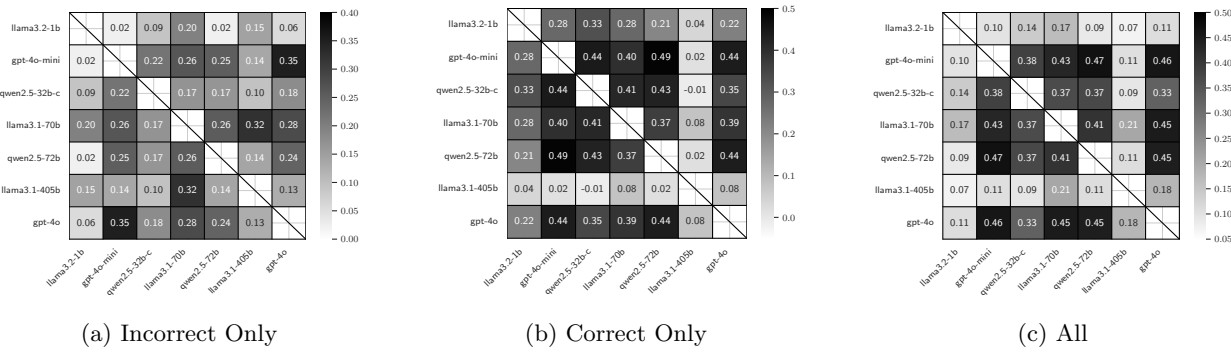

Figure 14: MedMCQA: Kendall's $\tau$ rank correlations of Llama3, GPT-4o, and Qwen2.5 models ordered by size.

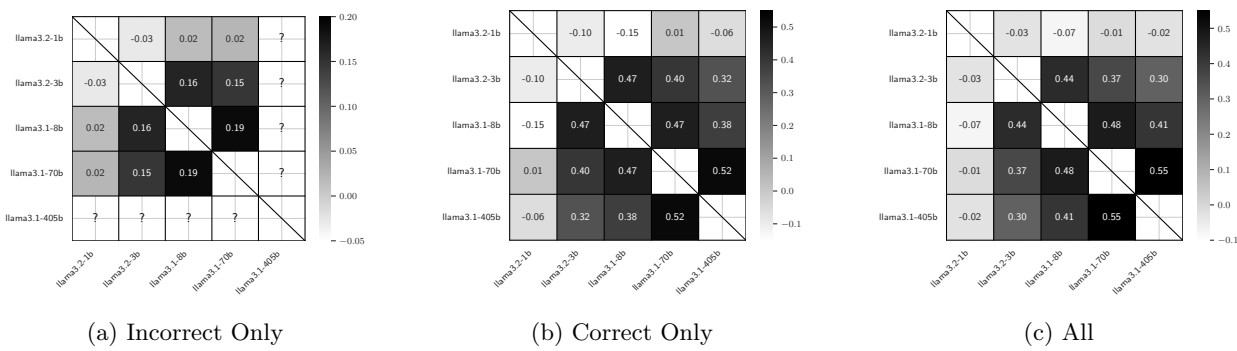

Figure 15: TriviaQA: Kendall's $\tau$ rank correlations of Llama3 models ordered by size.

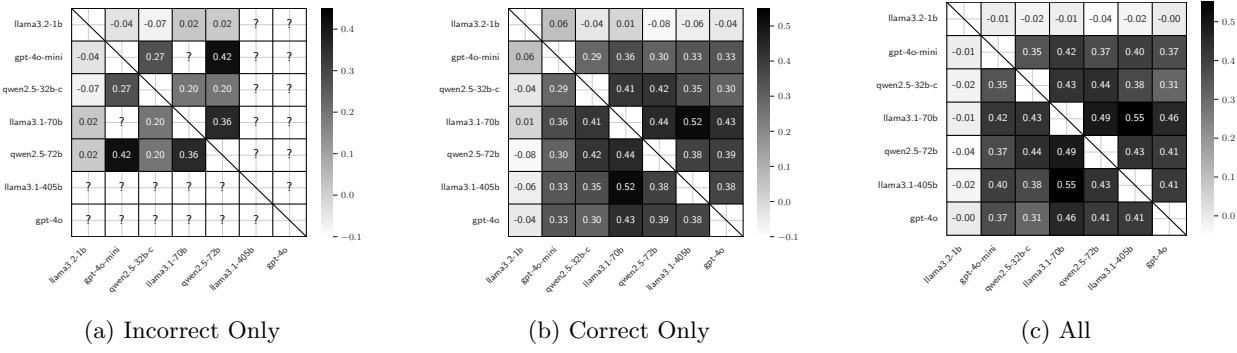

Figure 16: TriviaQA: Kendall's $\tau$ rank correlations of Llama3, GPT-4o, and Qwen2.5 models ordered by size.

correlations with sample size less than 50; we use the $n = 50$ cut-off since it reduces the standard error for Kendall's $\tau$ to around $\sigma_\tau \le 0.1$, based on a normal approximation.

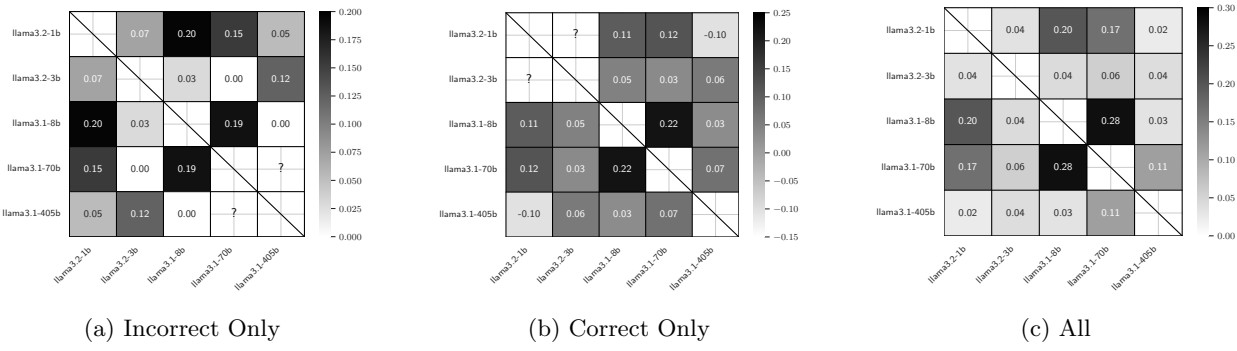

(a) Incorrect Only       (b) Correct Only       (c) All

Figure 17: XSum: Kendall's $\tau$ rank correlations of Llama3 models ordered by size.

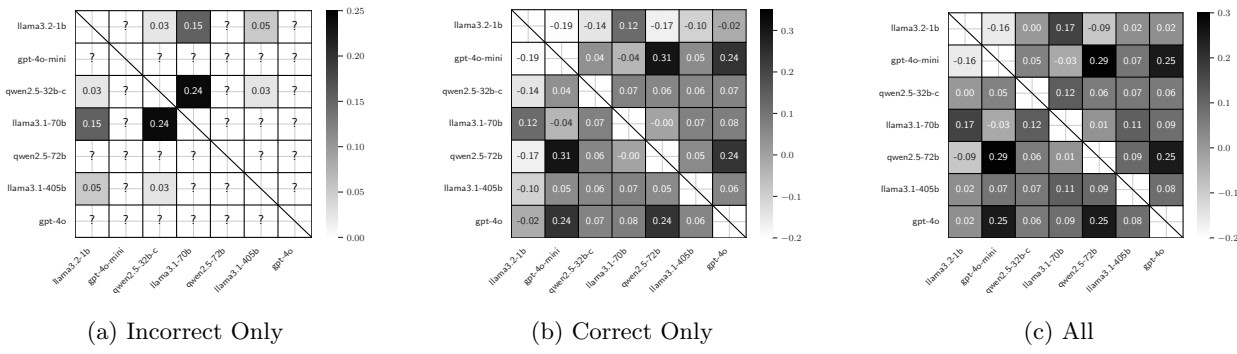

(a) Incorrect Only       (b) Correct Only       (c) All

Figure 18: XSum: Kendall's $\tau$ rank correlations of Llama3, GPT-4o, and Qwen2.5 models ordered by size.

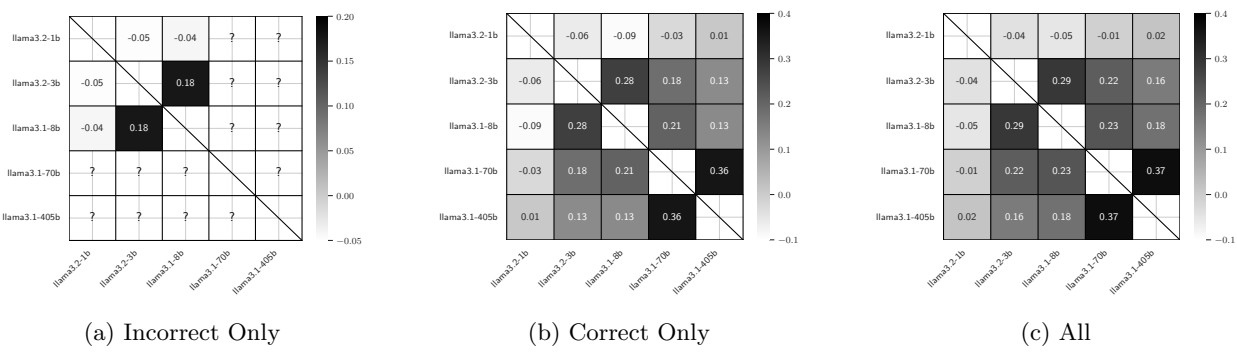

(a) Incorrect Only       (b) Correct Only       (c) All

Figure 19: GSM8K: Kendall's $\tau$ rank correlations of Llama3 models ordered by size.

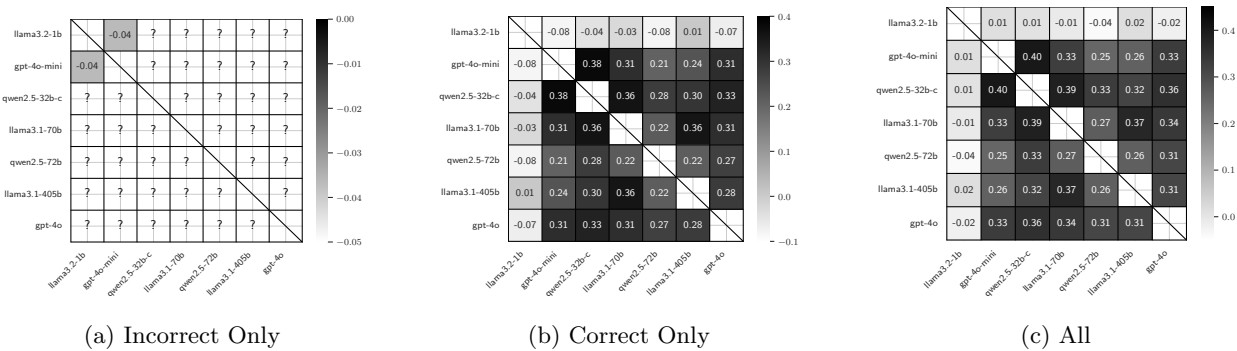

(a) Incorrect Only       (b) Correct Only       (c) All

Figure 20: GSM8K: Kendall's $\tau$ rank correlations of Llama3, GPT-4o, and Qwen2.5 models ordered by size.

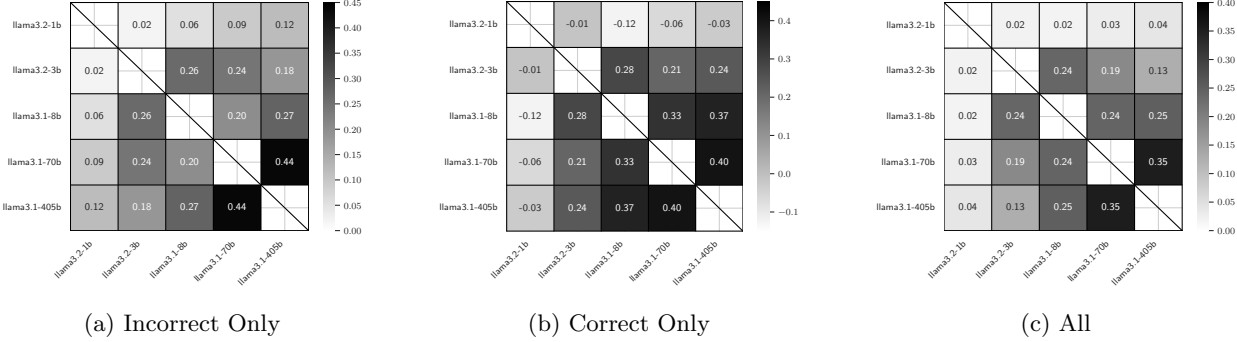

Figure 21: TruthfulQA: Kendall's $\tau$ rank correlations of Llama3 models ordered by size.

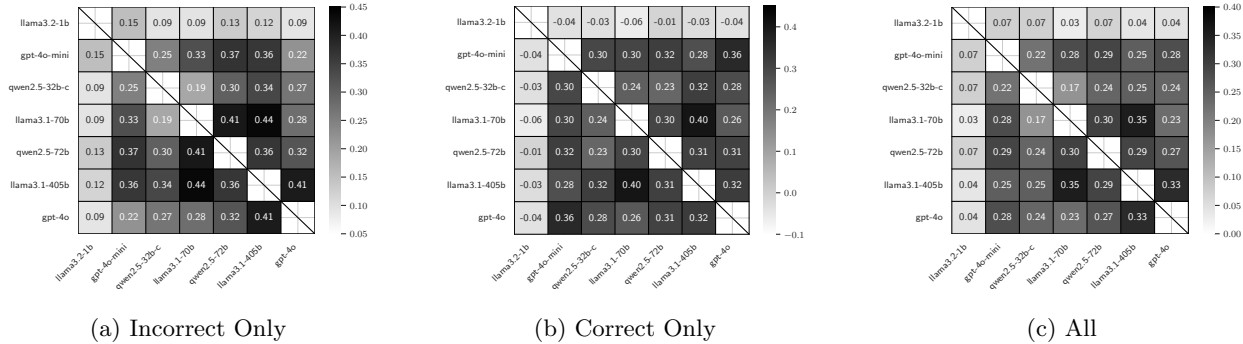

Figure 22: TruthfulQA: Kendall's $\tau$ rank correlations of Llama3, GPT-4o, and Qwen2.5 models ordered by size.

Table 10: Rank correlations of calibrated confidences between Llama3 1B, 3B, 8B, 70B, and 405B models, computed separately for *incorrectly* answered queries ("inc"), *correctly* answered queries ("corr"), and *all* queries ("all"): $\overline{\tau}$ is the average rank correlation across the 10 model pairs; $\tau_{a,b}$ is the rank correlation between data subsets (inc, corr, all) across model pairs; and $p_{a,b}$ is the $p$ value for $\tau_{a,b}$.

| Benchmark | $\overline{\tau}_{\text{inc}}$ | $\overline{\tau}_{\text{corr}}$ | $\overline{\tau}_{\text{all}}$ | $\tau_{\text{inc,corr}}$ | $p_{\text{inc,corr}}$ | $\tau_{\text{inc,all}}$ | $p_{\text{inc,all}}$ | $\tau_{\text{corr,all}}$ | $p_{\text{corr,all}}$ |
|---|---|---|---|---|---|---|---|---|---|
| **MMLU** | 0.199 | 0.396 | 0.385 | 0.505 | $< 0.0001$ | 0.717 | $< 0.0001$ | 0.686 | $< 0.0001$ |
| **MedMCQA** | 0.174 | 0.295 | 0.271 | 0.324 | 0.0055 | 0.486 | $< 0.0001$ | 0.730 | $< 0.0001$ |
| **TriviaQA** | 0.110 | 0.274 | 0.298 | 0.371 | 0.0014 | 0.413 | 0.0004 | 0.819 | $< 0.0001$ |
| **XSum** | 0.138 | 0.049 | 0.065 | 0.180 | 0.1284 | 0.187 | 0.1148 | 0.721 | $< 0.0001$ |
| **GSM8K** | 0.199 | 0.169 | 0.201 | 0.425 | 0.0003 | 0.467 | $< 0.0001$ | 0.921 | $< 0.0001$ |
| **TruthfulQA** | 0.222 | 0.196 | 0.178 | 0.667 | $< 0.0001$ | 0.860 | $< 0.0001$ | 0.756 | $< 0.0001$ |

