# OpenReview forum: "Rational Tuning of LLM Cascades via Probabilistic Modeling"
_TMLR — Accepted by TMLR_

### Review · Reviewer_6MPw · 2025-03-01

**Summary Of Contributions:**

This paper looks at how to make systems of LLMs work better together with cascades.

While confidence calibration is well-studied in ML, this work addresses the unique challenges of LLM cascades, where error rates interact dynamically across models.

The proposed model offers a fresh perspective by combining Markov assumptions (simplifying dependencies) with copulas (modeling pairwise correlations), enabling tractable optimization of thresholds.

**Audience:**

Yes

**Broader Impact Concerns:**

No major ethical or societal risks.

**Claims And Evidence:**

Yes

**Requested Changes:**

The text claims Proposition 2’s proof is in Appendix B, but it appears in Appendix A.

**Strengths And Weaknesses:**

Strength: Comprehensive experiments. Theoretical framework is logically sound for modeling cascade performance (up to Proposition 2).

The derivation of P(correct) includes a term P(Phi_1<=phi_1) even for i=1, which feels counterintuitive (why multiple  the first model’s low confidence probability when it is selected). The proof does not fully address the i=1 edge case either. Equation (26) appears to be wrong.

---

> ### Author Response · Authors · 2025-03-03
> **Response to Reviewer 6MPw: All Suggestions Implemented**
>
> Thank you for your thoughtful review. We are delighted that our work resonates with you.
>
> Indeed, Proposition 2 displayed the formulas for P(Correct) and E[Cost] incorrectly. We apologize for this oversight during proofreading. Our codebase already implemented the i=1 edge case correctly, and no change was required there. In our revised manuscript, we
> - fixed the presentation of Proposition 2
> - expanded the proof in Appendix A to explicitly treat the i=1 case
> - fixed the reference to the proof of Proposition 2, so that it correctly points to Appendix A rather than Appendix B
>
> Thank you again for your careful review! Your feedback is much appreciated. Please let us know if our revision meets your approval and if there are any additional changes you would recommend.

---

### Review · Reviewer_Z6N8 · 2025-03-17

**Summary Of Contributions:**

This paper proposed a probabilistic model for the joint distribution of calibrated confidence scores of LLMs in an LLM cascade. The LLM cascade in this context, describes a process where LLMs with growing performance (i.e. size) are solicited in sequence if the calibrated confidence level of the previous cascade member fails to reach a given threshold. The probabilistic model relies on several assumptions:
- 1. A parametric univariate distribution describing the marginal of each LLM
- 2. A Gumbel copula model to capture successive pairwise correlations.
- 3. The Markov property.

The probabilistic model is then used to tune the confidence thresholds by efficiently solving an optimization problem. The experiments include statistical tests to assess the correctness of the introduced assumptions, the individual performance and calibration of LLMs on standard benchmarks, and the comparison of the probabilistic model-enabled tuning of the cascade against a grid search baseline.

**Audience:**

Yes

**Broader Impact Concerns:**

There is no *Broader Impact Statement* in the paper, and no particular ethical considerations to raise.

**Claims And Evidence:**

Yes

**Requested Changes:**

Here is a summary of requested changes, of which details are given in the Weaknesses section above:
1. Analyzing the statistical validity of the assumptions in light of the final performance of the cascade.
2. Repeat the correlation experiments for each group (correct and incorrect answers) to avoid potential misleading interpretations.
3. Further quantify the LLM inference time gain from the improvement in the AUC.
4. Try hyperparameters search engines instead of the grid-search baseline.

**Strengths And Weaknesses:**

**Strengths:**

- The concept of LLM cascade is highly relevant in a time of LLM integration in various research areas and business applications.
- The idea of modeling the joint calibrated confidence scores is novel, and the underlying motivation of tuning the confidence scores of the LLM cascade is relevant for the community.
- Given the experiments, in selected benchmarks and cascade configurations (e.g. the cascade members belong to the same family of models), the assumptions underlying the proposed model are empirically validated on test-data.
- The proposed confidence levels tuning algorithm benefits from linear (in the number of LLMs in a cascade) time to compute the objective function, therefore, scaling better than the grid search baseline.

**Weaknesses:**

- Relevance of statistical validity of the assumptions to the final performance of the cascade: The results section includes many experiments to test the statistical validity of the probabilistic model's assumptions, these experiments showcase statistical significance levels, number of rejections of the null hypothesis, p-values, and all the relevant statistical measurements. However, the authors do not discuss how this impacts the performance of the underlying LLM cascades. For instance, one may wonder how sensitive the solution of your tuning algorithm is to the rejection rate of the null hypothesis in the CvM test of the Gumbel copula model.
- Potential confounding in the interpretation of the correlation experiments: In all of figures 1, 2, and 3 where the correlation scores between the calibrated confidence levels of pairwise LLMs are showcased, it seems that a strong correlation also correlates with similar performance (%Corr) in the respective benchmark. For instance, in Figure 1,c) the strongest correlation is between qwen2.5-72b and gpt-4o-mini who achieve respectively 69.1% and 66.% correctness in the corresponding benchmark. Similarly, in Figure 3,c) the strongly correlated llama3.1-405b and llama3.1-70b achieve respectively 81.5% and 85.2% in the corresponding benchmark. This suggests that the correlation observed in the confidence scores can be dominated by the correct answers that exhibit similar proportions whenever a strong correlation is observed. To make sure this is not the case, I suggest looking at the correlations between confidence scores grouped by correct and incorrect answers, to make sure both groups lead to similar conclusions.
- Unclear trade-offs between AUC improvement and actual computational cost: What does a 2% improvement in AUC translate to in terms of computational efficiency? For a fixed performance level, how much faster is LLM inference using your method compared to grid search?
- Possibly suboptimal grid-search baseline: Unless the grid search baseline is given enough budget to explore the full search space, other hyperparameters search engines might be able to find the optimal confidence thresholds in less time. Did the authors try that? This would also enable the search in the continuous space of thresholds (as opposed to the 2.5% resolution you currently use for grid search) as some engines parameterize distributions over continuous search spaces (e.g. HEBO https://arxiv.org/pdf/2012.03826).
- Lack of practical use case guidance: summing up some of the already stated weaknesses, how can the person interested in applying an LLM cascade optimally configure the cascade in light of your contributions? For user-chosen LLMs, what are the most crucial assumptions to validate to use of the proposed probabilistic model and tuning algorithm? Starting from which level of statistical invalidity of the assumptions does the grid search baseline become a better alternative? Are there any benchmark-related or task-related (multiple-choice questions, text generation) conclusions?

---

> ### Author Response · Authors · 2025-04-07
> **Response to Reviewer Z6N8: All Suggestions Implemented**
>
> Thank you for taking the time to craft such a careful review! We have implemented each of your suggestions as follows:
>
> - We added Section 4.4.2 discussing the sensitivity of our method’s performance gain (relative to high-resolution grid search) to the Cramér-von Mises (CvM) statistics assessing goodness-of-fit of the copulas and marginal distributions. The results are that performance appears to be more sensitive to the copula CvM statistic rather than the marginal CvM statistic. See Figure 8.
>
> - We adapted Section 4.4 to explain the implications of our AUC performance metric more clearly. Our revised manuscript emphasizes that the main goal of our Rational Tuning framework is to improve the trade-off between the error rate and inference cost (in dollars per query) of an LLM cascade. In other words, we aim to reduce the error rate at any cost level through optimal use of smaller (lower GPU dollar cost) vs larger (higher GPU dollar cost) models depending on the difficulty of the query. The newly added Figure 5 clarifies the computation of our AUC performance metric, and we have added the following intuitive interpretation: lowering the AUC by 1% means that the LLM cascade has a 1% lower error rate at any level of inference spend (dollars per query). Note that LLM inference speed (latency) is a separate issue. However, our framework can also be used to optimize latency instead of inference cost, by simply replacing the model-specific cost constants from dollars per query to the appropriate latency constants of milliseconds per query. Smaller transformers models typically have lower latency than larger transformers models, so ordering the models by inference cost is often equivalent to ordering them by latency.
>
> - We appreciate your suggestion to re-compute the rank correlations between LLMs’ confidence scores conditional on correct vs incorrect answers. We have added Appendix F providing detailed comparisons of rank correlation plots computed on “Correct Only”, “Incorrect Only”, and “All” examples. Figures 11-22 show that the rank correlations computed on “Correct Only” and “Incorrect Only” answers are generally similar to the overall rank correlations on "All" examples. Table 10 formalizes this claim by exhibiting the correlations between the rank correlation values conditioned on correct vs incorrect answers.
>
> - We added a Bayesian optimization baseline to Section 4.4. Following your suggestion, we evaluate our Rational Tuning framework against HEBO with the default Gaussian process surrogate function. The results are shown in Figure 6 and Table 7. Although HEBO performs impressively well, it generally performs worse than both our high-resolution grid search baseline and our Rational Tuning framework. Specifically, our method registers a 4.3% average performance improvement over HEBO for LLM cascades consisting of $k \geq 3$ models.
>
> - We appreciate your encouragement to provide practical guidance for applying our Rational Tuning framework. We added Section 4.4.4 presenting our central recommendation for practitioners: use one-dimensional grid search for two-model cascades with a single deferral threshold, and use Rational Tuning for longer cascades with multiple deferral thresholds. In addition, Section 4.4.4 suggests several visual diagnostics for verifying that the statistical assumptions underlying our probabilistic model approximately hold.
>
> In sum, we greatly appreciate your thoughtful engagement with our work. Please let us know if the changes we made in our revised manuscript fully address your suggestions. We look forward to hearing from you. Thank you again for your thorough review.

---

> > ### Comment · Reviewer_Z6N8 · 2025-04-17
> > **Thank You**
> >
> > I thank the author for all the additional experiments and clarifications that address my concerns and improve the manuscript quality. I recommend acceptance and appreciate all the authors' efforts.
> >
> > Best,
> >
> > Reviewer Z6N8

---

### Review · Reviewer_jx1j · 2025-03-23

**Summary Of Contributions:**

This paper presents a novel probabilistic framework for tuning the confidence thresholds of Large Language Model (LLM) cascades. By modeling the calibrated confidences of LLM with a Markov factorization and Gumbel copulas, the authors derive continuous, differentiable expressions for both the probability of correctness and expected cost. They then use these expressions in a continuous optimization procedure to select confidence thresholds that minimize error at a given cost. Empirical results across multiple benchmarks demonstrate its efficacy.

**Audience:**

Yes

**Claims And Evidence:**

Yes

**Requested Changes:**

N/A

**Strengths And Weaknesses:**

**Strengths**
- Comprehensive probabilistic framework that handles multi-model error correlation using copulas.
- Continuous-optimization approach provides better scalability than grid search, crucial for longer cascades.
- Goodness-of-fit experiments on six benchmarks and multiple LLMs demonstrate the adequacy of the model.
- Includes careful calibration and ablation to reduce overconfidence.
- Empirical results show robust improvements in cost-error tradeoffs, highlighting practical value.

**Weakness**
- The paper’s approach relies on labeled data (300 training examples) to calibrate confidences; in settings with very limited data, performance might degrade.

---

> ### Author Response · Authors · 2025-04-07
> **Response to Reviewer jx1j: All Suggestions Implemented**
>
> Thank you for your thoughtful review. We appreciate your encouragement to examine the performance of our Rational Tuning framework in the low-sample regime.
>
> Following your suggestion, we added Section 4.4.1 discussing the performance of our methodology with $n \leq 30$ training examples. The results are highly favorable: our Rational Tuning framework outperforms Bayesian optimization by up to 16.5% and high-resolution grid search by up to 2.7%. See Section 4.4.1, Figure 7, and Table 8. These results lead us to conclude that the inductive bias of our probabilistic model enhances sample efficiency.
>
> Thank you again for your thoughtful feedback on our work. Your engagement has significantly improved the quality of our manuscript.

---

> > ### Comment · Reviewer_jx1j · 2025-04-22
> > **Thank you**
> >
> > I thank the authors for all the additional clarifications that address my concerns and improve the manuscript quality.
> >
> > Best,
> > Reviewer jx1j

---

### Decision · Action_Editor_scJ9 · 2025-05-16

**Recommendation:** Accept as is

**Comment:**

Reviewers were unanimously positive in their assessment of the paper. There were some suggestions regarding the experiments, such as considering smaller sample sizes, better discussing the relation of AUC gains to final cascade performance, and more careful analysis and discussion of the statistical significance. These were satisfactorily addressed by the authors in the response.

We have two minors suggestion that the authors may optionally consider:
- you may wish to defer some of the experimental results, and discussion thereof, to the Appendix; in the current form, the main body runs a little long.
- as noted in Section 4.4.4, the benefit of the framework could be in settings with multiple models. One challenge with cascades of several models is that the incurred cost could be non-trivial, as it is the cumulative cost of all models invoked. This is in contrast to simple query routing, which makes a one-shot model selection without invoking multiple models. This may be something worth discussing (for future work).

Overall, the paper makes an interesting new contribution to a problem of practical importance, with both mathematical and empirical analysis that could be built on by future work. We thus recommend acceptance for publication at TMLR.

**Audience:**

The study of model cascades is of growing importance given their potential to lower inference costs. Advances in the understanding and modelling of such cascades is thus expected to be of interest to a good portion of the TMLR audience.

**Claims And Evidence:**

The paper's central claims are that:
* to study the behaviour of model cascades, one may tractable model the joint distribution of the constituent model confidences via a Marko-copula model
* building on this, one may identify suitable confidence thresholds by exploiting analytical formulae for the cost-quality tradeoffs
* the resulting thresholds perform favourably compared to standard tuning

The first two claims are backed by mathematical derivations that are clearly laid out. The third claim is backed by extensive experiments on a family of different models across multiple benchmarks.

---

> ### Author Response · Authors · 2025-06-06
> **Response to Action Editor**
>
> Thank you very much for the acceptance decision and your optional suggestions.
>
> We carefully considered your suggestions. While we've retained the experimental results in the manuscript's main body, we included a new paragraph in the conclusion to indicate our Rational Tuning framework’s applicability to the one-shot routing problem. Specifically, the new passage reads:
> > “Building on these promising results, an interesting direction would be to apply our probabilistic modeling framework to LLM routing, in which a central routing model sends a query to the most suitable LLM in a single step, avoiding cumulative cost increases as the query propagates down a cascade. Since cumulative cost increases are especially severe for longer cascades (at which our methodology excels), the routing setting may more effectively leverage Rational Tuning's capacity for modeling dependencies between arbitrarily many distinct LLMs. For instance, suppose the routing decision depends on noisy estimates $\hat{\phi}_1, ..., \hat{\phi}_n$ of the LLMs' true calibrated confidences. In this case, balancing the noisy observations $\hat{\phi}_i$ against their probabilistic expectations $\mathbb{E}[\hat{\phi}_i | \hat{\phi}_1, ..., \hat{\phi}_{i-1}, \hat{\phi}_{i+1}, ... \hat{\phi}_n]$ may lead to more effective routing decisions.”
>
> We greatly appreciate your thoughtful feedback.